# The solute carrier SLC9C1 is a Na$^+$/H$^+$-exchanger gated by an S4-type voltage-sensor and cyclic-nucleotide binding

F. Windler[1,2], W. Bönigk[1], H.G. Körschen[1], E. Grahn[1], T. Strünker[1,2,3], R. Seifert[1,2] & U.B. Kaupp[1,2,4]

Voltage-sensing (VSD) and cyclic nucleotide-binding domains (CNBD) gate ion channels for rapid electrical signaling. By contrast, solute carriers (SLCs) that passively redistribute substrates are gated by their substrates themselves. Here, we study the orphan sperm-specific solute carriers SLC9C1 that feature a unique tripartite structure: an exchanger domain, a VSD, and a CNBD. Voltage-clamp fluorimetry shows that SLC9C1 is a genuine Na$^+$/H$^+$ exchanger gated by voltage. The cellular messenger cAMP shifts the voltage range of activation. Mutations in the transport domain, the VSD, or the CNBD strongly affect Na$^+$/H$^+$ exchange, voltage gating, or cAMP sensitivity, respectively. Our results establish SLC9C1 as a phylogenetic chimaera that combines the ion-exchange mechanism of solute carriers with the gating mechanism of ion channels. Classic SLCs slowly readjust changes in the intra- and extracellular milieu, whereas voltage gating endows the Na$^+$/H$^+$ exchanger with the ability to produce a rapid pH response that enables downstream signaling events.

[1] Center of Advanced European Studies and Research (caesar), Department Molecular Sensory Systems, Ludwig-Erhard-Allee 2, 53175 Bonn, Germany. [2] Marine Biological Laboratory, 7 MBL Street, Woods Hole 02543 MA, USA. [3] University Hospital Münster, Center of Reproductive Medicine and Andrology, Albert-Schweitzer-Campus 1, Geb. D11, 48149 Münster, Germany. [4] University of Bonn, Life & Medical Sciences Institute (LIMES), Carl-Troll-Str. 31, 53115 Bonn, Germany. Correspondence and requests for materials should be addressed to R.S. (email: r.seifert@caesar.de) or to U.B.K. (email: u.b.kaupp@caesar.de)

Solute carriers (SLC), only second to GPCRs, form one of the largest gene families in vertebrates, comprising about 450 members in the human genome. Yet, compared to other gene families, SLCs are understudied and many isoforms represent orphan proteins, highlighting our ignorance[1]. A case in point is the subfamily SLC9C1, also referred to as sNHE. SLC9C1 has been suspected to serve as Na+/H+ exchanger that controls intracellular pH (pH$_i$) in mammalian sperm[2].

Changes in pH$_i$ are key to sperm signaling[3–11], but it is not known if SLC9C1 indeed promotes Na+/H+ exchange, how its activity is controlled, and whether it contributes to pH$_i$ regulation. Disrupting the mouse *slc9c1* gene renders sperm immotile and male mice infertile[2,12], demonstrating that SLC9C1 is required for sperm function and fertilization in mammals. However, the pH$_i$ of SLC9C1$^{-/-}$ sperm is not altered[2], and a clear-cut conclusion is compounded by the unexpected observation that cAMP synthesis is impaired in SLC9C1$^{-/-}$ sperm and that the motility defect can be rescued by cAMP[12,13]. These observations suggest that the prime defect of SLC9C1$^{-/-}$ sperm might be in cAMP—rather than pH$_i$ signaling. Finally, mouse SLC9C1 is non-functional in heterologous systems and attempts to study SLC9C1-mediated Na+/H+ exchange by pH$_i$ fluorimetry in mouse sperm were unsuccessful[2]. Thus, the function of SLC9C1 as a Na+/H+ exchanger and its role in pH$_i$ regulation of mammalian sperm is left in limbo.

On a different yet related note, in sea urchin sperm, chemoattractants stimulate a rapid rise of pH$_i$[9,14–20], which serves as a switch to activate the pH-sensitive CatSper Ca$^{2+}$ channel that controls chemotaxis[9]. The molecule underlying this alkalinization is not known. However, a Na+/H+ exchange mechanism was described that is activated by hyperpolarization rather than by changes in the extracellular pH or Na+ concentrations[14,15,17]. Ever since its discovery, the activation of Na+/H+ exchange by voltage and the underlying mechanism and molecules have remained unexplained.

We report here final success in solving these fundamental and long-standing questions. We demonstrate that the *Strongylocentrotus purpuratus* homolog (*Sp*SLC9C1) exists in sperm and represents a genuine Na+/H+ antiporter. Unlike solute carriers, Na+/H+ exchange by *Sp*SLC9C1 is gated by voltage via a voltage-sensing domain (VSD) and directly modulated by cAMP via a cyclic nucleotide-binding domain (CNBD). Thus, we deorphanize the SLC9C1 family and identify Na+/H+ exchange as a target for cAMP signaling and a mechanism of adaptive interaction between pH$_i$ and cAMP. On a broader perspective, our results now enable future studies of the commonalities and differences of voltage sensing and cAMP modulation between ion channels and a solute carrier and, thereby, gain insight into the evolution of gating mechanisms.

## Results

**Overall protein topology of SLC9C1.** Sperm-specific Na+/H+ exchangers share with SLC9 family members the exchanger domain that carries substrates across membranes. In addition, SLC9C1 holds a putative voltage-sensing domain (VSD) and a putative cyclic nucleotide-binding domain (CNBD) that are absent in other SLC9 members (Fig. 1a). The exchange domain of *Sp*SLC9C1 is predicted to encompass 14 transmembrane segments (TM), whereas bacterial and archaeal Na+/H+ exchangers feature 12 and 13 TMs, respectively[21–23]. A sequence alignment illustrates that *Sp*SLC9C1 carries an additional TM at the N-terminal end (Fig. 1a and Supplementary Figure 1). The Na+-binding site in archaeal SLC9 forms a trigonal bipyramid[22,23]. In *Methanocaldococcus jannaschii* NhaP1, two carboxyl groups (D132 and D161) and a hydroxyl group (S157) form

a triangle around Na+; the main-chain carbonyl of T131 is positioned at one bipyramid tip and T76/E154 at the other tip (Fig. 1b). All but one of the important Na+-coordinating residues are conserved in *Sp*SLC9C1, including an Asn/Asp motif (Asn237/Asp238) that is diagnostic for electroneutral exchangers[24] (Fig. 1a, b). Residues occupying the (T76/E154) pyramid tip are less conserved, even among archaeal SLCs. Furthermore, two Arg residues in TMs 12 and 13 that are functionally important in other Na+/H+ exchangers are conserved in *Sp*SLC9C1 (Fig. 1a and Supplementary Figure 1).

The full-fledged VSD (four transmembrane domains S1–S4) of *Sp*SLC9C1 carries seven conserved Arg or Lys residues in the S4 motif and four conserved Glu or Asp residues in S1–S3 (Fig. 1c). Finally, the CNBD features the hallmarks of a cyclic nucleotide-binding fold: three α-helices (αA, αB, and αC), eight β-strands (β1–β8), and a phosphate-binding cassette (PBC) (Fig. 1d). Key residues that interact with cyclic nucleotides are conserved, including the purine-binding residues Val and Leu (β4 and β5), the ribofuranose-binding residues Gly/Glu (β6), the phosphate-binding Arg between β6 and β7, and the purine-binding residues Arg/Lys in αC[25,26]. The presence of exchanger, VSD, and CNBD domains suggests that SLC9C1 promotes Na+/H+ exchange controlled by voltage and cyclic nucleotides. We studied by heterologous expression of *Sp*SLC9C1 its voltage sensitivity, Na+/H+ exchange activity, and regulation by cyclic nucleotides.

**The voltage-sensing domain produces gating currents.** For electrophysiological experiments, we used CHO cells stably expressing HA-tagged *Sp*SLC9C1. An anti-HA antibody stained sheets of plasma membrane (Supplementary Figure 2), showing that *Sp*SLC9C1 reaches the cell membrane. We tested by whole-cell patch-clamping whether the VSD is functional and displays charge movements during voltage steps. In fact, several different ion channels, e.g., CNG channels, carry a VSD, yet are not gated by voltage and are not voltage-dependent. In voltage-activated ion channels, the movement of charged amino acids in S4 during activation produces so-called gating currents[27]. Brief voltage pulses (−15 to −155 mV) evoked transient negative and positive gating currents at the onset and termination of the voltage pulse, respectively (Fig. 2a). In control cells, voltage steps did not evoke gating currents (Fig. 2a). A fit of the Boltzmann function to the integrated off-gating currents yielded a voltage of half-maximal activation ($V_{1/2}$) and slope factor ($s$) of −94.7 ± 2.9 and 8.5 ± 0.8 mV, respectively, corresponding to a gating charge $q_g$ of 3.1 $e_o$ (Fig. 2a, b and Table 1). In conclusion, the VSD in *Sp*SLC9C1 is functional.

In voltage-gated K+ channels, substituting Arg residues in the S4 motif for neutral residues shifts $V_{1/2}$ to more negative values[28–32]. Replacing the third Arg residue in the S4 segment of *Sp*SLC9C1 by Gln (R803Q) shifted the $V_{1/2}$ of gating-current activation by −24 to −117.9 ± 7.1 mV (Fig. 2c and Supplementary Figure 4a); concomitantly, the number of gating charges was lowered to 2.0 $e_o$ (Fig. 2c and Table 1). Arg803 apparently contributes one equivalent gating charge, indicating that it may cross the entire transmembrane electric field, similar to the homologous Arg368 in *Shaker* K+ channels[32].

***Sp*SLC9C1 mediates voltage-gated Na+/H+ exchange.** No currents other than gating currents were observed, indicating that Na+/H+ exchange is electroneutral. We tested Na+/H+ exchange activity of *Sp*SLC9C1 in the whole-cell configuration by voltage-clamp fluorimetry, using the pH indicator BCECF. For inwardly directed Na+ gradients, stepping $V_m$ from −40 to −100 mV enhanced the BCECF fluorescence ratio $R$ ($F_{480}/F_{440}$), indicating

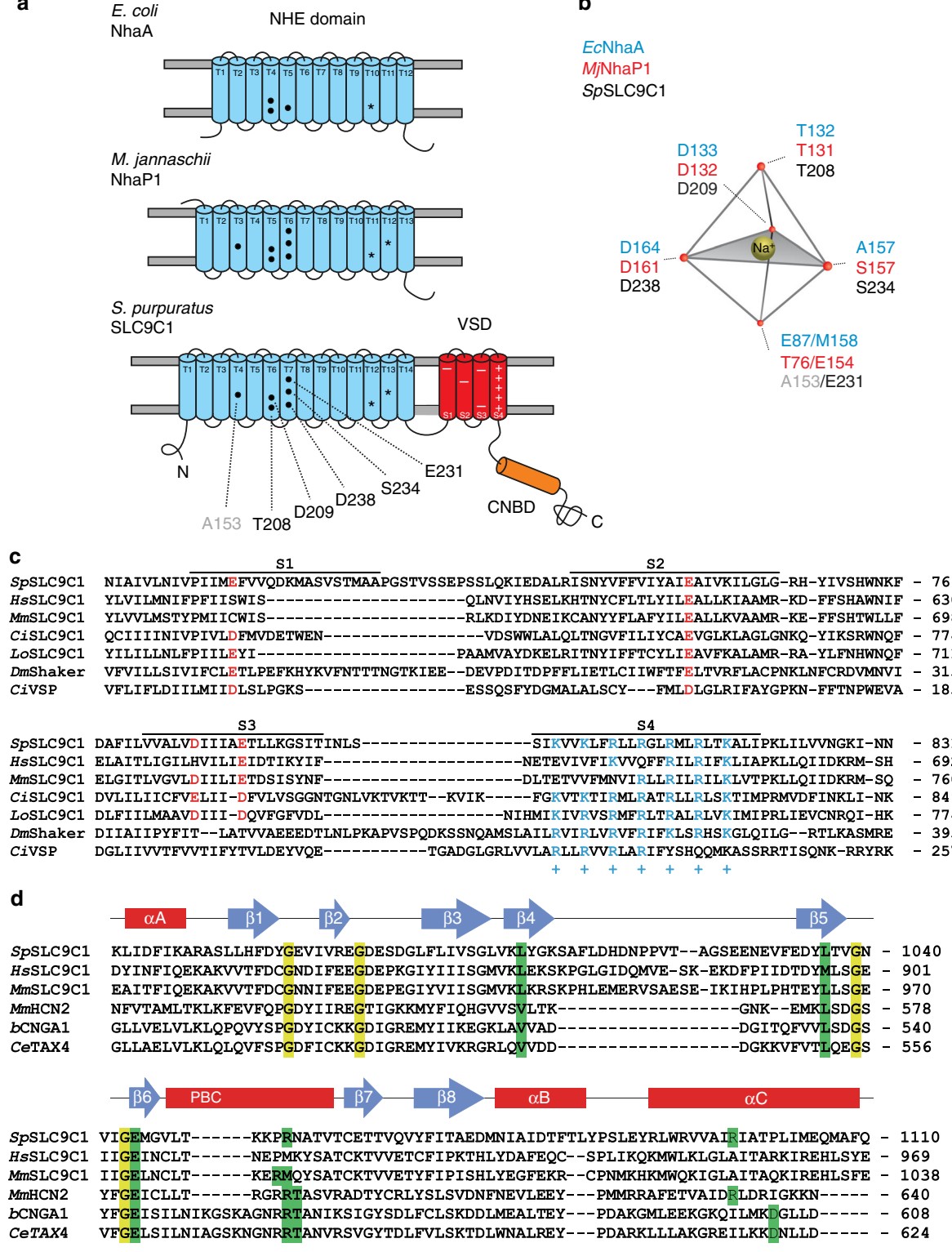

intracellular alkalinization (Fig. 3a). After the voltage was stepped back, exchange activity stopped and pH$_i$ slowly returned to baseline (Fig. 3a). In non-transfected cells, this voltage protocol did not change pH$_i$, showing that potential endogenous Na$^+$/H$^+$ exchangers do not operate under these conditions (Fig. 3a). Additional controls ascertain that the changes in pH$_i$ result from

Na$^+$/H$^+$ exchange of SpSLC9C1: first, when extracellular Na$^+$ was substituted for NMDG, i.e., when the Na$^+$ gradient was reversed, the fluorescence ratio decreased upon hyperpolarization, reflecting intracellular acidification (Fig. 3b). Second, when both pH and [Na$^+$] are symmetric across the membrane, stepping $V_m$ to −100 mV did not change pH$_i$ (Fig. 3c). The suppression was

**Fig. 1** Structural features of *Sp*SLC9C1. **a** Transmembrane topology of *Escherichia coli* NhaA (*Ec*NhaA), *Methanocaldococcus jannaschii* NhaP1 (*Mj*NhaP1), and sea urchin *Strongylocentrotus purpuratus* SLC9C1 (*Sp*SLC9C1); VSD voltage-sensing domain, CNBD cyclic nucleotide-binding domain. Amino acids that may participate in Na$^+$ coordination are highlighted (black dots). Asterisks indicate conserved Arg residues that are relevant for Na$^+$/H$^+$ antiport; Arg320 and Arg347 in *Mj*NhaP1; Arg399 and Arg431 in *Sp*SLC9C1. An Arg is substituted by Lys300 in *Ec*NhaA. **b** Scheme of the trigonal bi-pyramidal structure of the Na$^+$ coordination site from *Mj*NhaP1. Numbers refer to the respective amino-acid residues in *E. coli* (blue), *M. jannaschii* (red), and *S. purpuratus* (black). **c** Sequence comparison of the VSD from several SLC9C1 members with the canonical VSD of *Drosophila* Shaker K$^+$ channel (*Dm*Shaker) and *Ciona intestinalis* voltage-sensor-containing-phosphatase (*Ci*VSP). *S. purpuratus* (*Sp*SLC9C1), *H. sapiens* (*Hs*SLC9C1), *M. musculus* (*Mm*SLC9C1), *C. intestinalis* (*Ci*SLC9C1), and spotted gar *L. oculatus* (*Lo*SLC9C1). Voltage sensors carry conserved positively charged residues in S4 (blue) and conserved negatively charged amino acids in S1–S3 (red). **d** Cyclic nucleotide-binding domains from sea urchin (*Sp*SLC9C1), human (*Hs*SLC9C1) and mouse (*Mm*SLC9C1) SLC9C1, mouse HCN channel *Mm*HCN2, bovine CNG channel *b*CNGA1, and *C. elegans* CNG channel (*Ce*TAX4). The CNBD comprises three α-helices (αA, αB, and αC), eight β-strands (β1–β8), and a phosphate-binding cassette (PBC). Highlighted key residues are the purine-binding residues Val and Leu (β4 and β5), the ribofuranose-binding residues Gly/Glu (β6), the phosphate-binding residues Arg/Thr, and the purine-binding Arg in αC of *Mm*HCN2

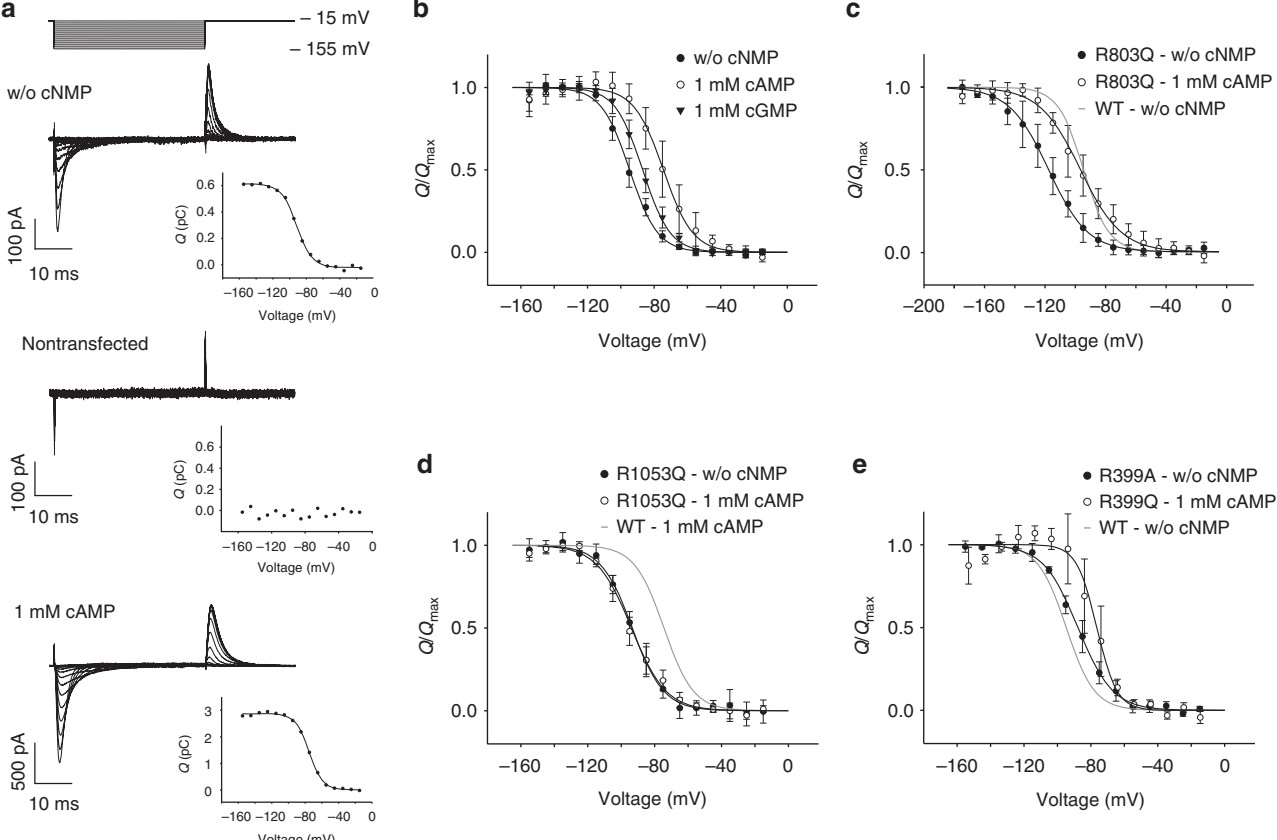

**Fig. 2** Gating currents of *Sp*SLC9C1. Voltage protocol, gating currents, and charge–voltage ($Q/V$) relation. The solid curve in the $Q/V$ relations (insets) represents a Boltzmann fit. **a** Upper: wt *Sp*SLC9C1 without cNMP ($V_{1/2} = -91.4$ mV, slope $s = 9.2$ mV). Middle: non-transfected CHO cells. Lower: wt *Sp*SLC9C1, 1 mM cAMP in the pipette solution ($V_{1/2} = -75.4$ mV, $s = 8.2$ mV). **b** $Q/Q_{max}$ vs. $V_m$ relation. Mean ± SD ($n$ = number of experiments) of $V_{1/2}$ and slope $s$ was determined by a Boltzmann fit (w/o cNMP: $-94.7 ± 2.9$ mV, $s = 8.5 ± 0.8$ mV, $q_g = 3.1$ $e_0$, $n = 6$; cAMP: $-74.4 ± 6.4$ mV, $s = 8.8 ± 1.9$ mV, $q_g = 2.9$ $e_0$, $n = 7$; cGMP: $-86.9 ± 3.0$ mV, $s = 8.3 ± 1.0$ mV, $q_g = 3.1$ $e_0$, $n = 7$). **c** Replacing Arg803 in S4 by Gln (R803Q) shifted $V_{1/2}$ by $-24$ mV and $s$ by 5 mV ($V_{1/2} = -117.9 ± 7.1$ mV, $s = 13.0 ± 1.1$ mV, $q_g = 2.0$ $e_0$, $n = 7$). This mutation did not affect $V_{1/2}$ modulation by 1 mM cAMP ($V_{1/2} = -96.8 ± 6.6$ mV, $s = 13.5 ± 2.6$ mV, $q_g = 1.9$ $e_0$, $n = 6$). Gray line: voltage dependence of wt *Sp*SLC9C1 without cNMP. **d** Replacing Arg1053 in the CNBD by Gln (R1053Q) does not affect $V_{1/2}$ without cNMP ($-93.4 ± 1.7$ mV, $s = 9.4 ± 2.0$ mV, $q_g = 2.8$ $e_0$, $n = 6$), but strongly reduced $V_{1/2}$ shift by cAMP ($V_{1/2} = -93.9 ± 4.2$ mV, $s = 10.3 ± 1.0$ mV, $q_g = 2.5$ $e_0$, $n = 5$). Gray line: voltage dependence of wt *Sp*SLC9C1 with cNMP. **e** In the NHE-domain mutant (R399A), Na$^+$/H$^+$ exchange is abolished (Fig. 5), but not gating currents (Supplementary Figure 4). $V_{1/2}$ in the absence ($-87.3 ± 2.8$ mV, $s = 10.4 ± 1.8$ mV, $q_g = 2.5$ $e_0$, $n = 3$) and presence of cAMP ($-76.5 ± 9.6$ mV, $s = 6.0 ± 0.8$ mV, $q_g = 4.3$ $e_0$, $n = 5$). Gray line: voltage dependence of wt *Sp*SLC9C1 without cNMP

reversible: when the Na$^+$ gradient was restored, hyperpolarization again evoked alkalinization (Fig. 3c). Furthermore, we compared the pH$_i$ buffer capacity and resting pH$_i$ (pH$_{rest}$) of control CHO- and CHO-*Sp*SLC9C1 cells using the pH$_i$ pseudo-null-point method[9,33–36]. The pH$_i$ changes imposed by the respective

pH$_i$ pseudo-null-point solutions were similar in control CHO- and CHO-*Sp*SLC9C1 cells (Supplementary Figure 3a, b), showing that expression of *Sp*SLC9C1 per se neither affects the pH$_i$ and buffering capacity of cells nor the kinetics and magnitude of the pH-indicator response.

**Table 1 Summary of activation properties of SpSLC9C1 from gating currents and voltage-clamp fluorimetry**

| | Gating currents | | | | Exchange activity | | |
|---|---|---|---|---|---|---|---|
| | $V_{1/2}$ (mV) | s (mV) | $N_q$ | n | $V_{1/2}$ (mV) | s (mV) | n |
| WT—w/o cNMP | −94.7 ± 2.9 | 8.5 ± 0.8 | 3.1 | 6 | −70.9 ± 2.5 | 3.3 ± 0.9 | 7 |
| WT—cAMP | −74.4 ± 6.4 | 8.8 ± 1.9 | 2.9 | 7 | −56.8 ± 2.7 | 4.6 ± 1.5 | 7 |
| WT—cGMP | −86.9 ± 3.0 | 8.3 ± 1.0 | 3.1 | 7 | −67.8 ± 5.4 | 2.8 ± 1.0 | 9 |
| R1053Q—w/o cNMP | −93.4 ± 1.7 | 9.4 ± 2.0 | 2.8 | 6 | −70.8 ± 4.8 | 3.6 ± 1.1 | 5 |
| R1053Q—cAMP | −93.9 ± 4.2 | 10.3 ± 1.0 | 2.5 | 5 | −69.5 ± 3.8 | 3.9 ± 1.2 | 6 |
| R803Q—w/o cNMP | −117.9 ± 7.1 | 13.0 ± 1.1 | 2.0 | 7 | n.d. | n.d. | |
| R803Q—cAMP | −96.8 ± 6.6 | 13.5 ± 2.6 | 1.9 | 6 | n.d. | n.d. | |
| R399A—w/o cNMP | −87.7 ± 2.8 | 10.4 ± 1.8 | 2.5 | 3 | n.d. | n.d. | |
| R399A—cAMP | −76.5 ± 9.4 | 6.0 ± 0.8 | 4.3 | 5 | n.d. | n.d. | |
| WT—caged cAMP | n.d. | n.d. | n.d. | | −55.3 ± 5.1 | 5.0 ± 0.6 | 3 |
| WT—500 μM Amiloride | n.d. | n.d. | n.d. | | −73.3 ± 2.2 | 2.8 ± 1.1 | 4 |

$V_{1/2}$ refers to the potential where $\Delta Q(V) = Q_{max}/2$ or $\Delta R (V) = \Delta R_{max}/2$ and s refers to the slope of the Boltzmann fit; n is the number of experiments and $N_q$ the number of charges involved in the gating process (the calculation is described in the Methods section). All values are given as mean ± SD

The NhaP1 of *M. jannaschii* harbors two essential Arg residues in transmembrane segments T11 (R320) and T12 (R347) (Supplementary Figure 1)[24]. When these residues in NhaP1 are mutated to Ala, the exchange activity is abolished (R320A) or severely impaired (R347A)[24]. We replaced in SpSLC9C1 the Arg residue, which is homologous to R320, by an Ala residue (R399A in TM12 (Supplementary Figure 1)). The R399A mutant produced gating currents with a $V_{1/2}$ and steepness s similar to those of the wild-type protein (Fig. 2e and Supplementary Figure 4b). However, Na+/H+ exchange activity was abolished in the R399A mutant (Fig. 3d). This result shows that in the R399A mutant transport activity of the exchanger domain is abolished without affecting the gating mechanism. In conclusion, SpSLC9C1 is a genuine Na+/H+ exchanger that, like other ion exchangers, can operate in the forward and reverse mode.

**Voltage dependence of exchange activity**. We determined the voltage dependence of SpSLC9C1 by recording Na+/H+ exchange activity at different voltages from one and the same cell. However, after stepping the voltage back to $V_{hold}$, the pH$_i$ returned only slowly to resting levels (Fig. 3a, b). Moreover, pH$_i$ signals started to run down during repetitive SpSLC9C1 activation, which prevented recording a complete voltage-response relation under stable conditions (Fig. 3e). To overcome these limitations, we co-expressed SpSLC9C1 with the proton channel Hv1 and recorded SpSLC9C1 activity in the reverse mode. The rationale underlying this strategy was: in the reverse mode, SpSLC9C1 activity acidifies the cell (Fig. 3b); subsequently, opening of Hv1 channels causes H+ efflux and restores the original pH$_i$. This prediction is borne out by experiments; upon hyperpolarization, SpSLC9C1 activity acidifies the cell, and activation of H+ outward currents via Hv1 by a subsequent depolarization[37] hastens the recovery from acidification (Fig. 3f). Because the operative voltage regimes of Hv1 and SpSLC9C1 do not overlap, H+ efflux via Hv1 does not interfere with H+ import via SpSLC9C1. SpSLC9C1 is activated at $V_m < −50$ mV, whereas for the pH gradients used here, Hv1 opens at $V_m \geq 0$ mV[38]. Furthermore, Hv1 only supports H+ outward currents[38]. In CHO cells expressing only Hv1, stepping $V_m$ from values between −23 and −113 mV to +47 mV produced a rapid Hv1-mediated increase of pH$_i$, whereas stepping voltage from +47 mV to negative values did not change pH$_i$ (Supplementary Figure 3c).

When SpSLC9C1 and Hv1 were co-expressed, the voltage protocol started with Hv1 activation to produce an initial alkalinization. This protocol provides a larger dynamic range of the BCECF dye, and the initial pH$_i$ value from which the

SpSLC9C1-mediated acidification started was always identical. In addition, this activation protocol ensures that the voltage dependence of Hv1 does not overlap with that of the SpSLC9C1 protein, because alkalinisation shifts the voltage dependence of Hv1 activation to more positive voltages. Repetitive Hv1 activation (for 10 s at about +50 mV) followed by SpSLC9C1 activation (for 10 s at about −100 mV) produced sawtooth-like cycles of alkalinization and acidification that were highly reproducible (Fig. 3f). Therefore, SpSLC9C1 activity at different voltages can be compared quantitatively. The voltage dependence of pH$_i$ responses was determined by stepping $V_m$ from a holding potential of +47 mV to values from −23 to −103 mV (Fig. 3g). A Boltzmann function fitted to the initial slope $\Delta R\ s^{-1}$ vs. $V_m$ yielded a mean $V_{1/2}$ of −70.9 ± 2.5 mV and slope factor $s = 3.3 ± 0.9$ mV (Fig. 3h and Table 1). The pH$_i$ responses were similar when the $V_m$ protocol was reversed (Supplementary Figure 3d).

We tested several generic inhibitors of Na+/H+ exchangers present in somatic cells[39]. None of these drugs affected the voltage-gated Na+/H+ exchange activity or voltage dependence of SpSLC9C1 activation (Supplementary Figure 5a, b, c, d), most likely because the sequence motifs and regions to which these NHE inhibitors bind[40] are either lacking or different in SpSLC9C1. The insensitivity of SpSLC9C1 to amiloride provided an opportunity to confirm that endogenous Na+/H+ exchangers in CHO cells are silent under our measuring conditions. An acid-load experiment using control CHO cells[41] (Supplementary Figure 6a) revealed endogenous amiloride-sensitive Na+/H+ exchange; however, this exchange does not interfere with voltage-gated SpSLC9C1 activity (Supplementary Figure 6b, c): the voltage dependence of SpSLC9C1 was similar in the absence and presence of amiloride (500 μM). In principle, SpSLC9C1 could exhibit resting activity at depolarized voltages, similar to CNG channels that have been shown to have a non-zero open probability in the absence of ligand[42]. We tried to estimate the resting activity of SpSLC9C1 at depolarized voltages (−30 mV) in the presence of amiloride (500 μM) by exchanging solutions from symmetrical with respect to Na+ and H+ to asymmetrical. Under these conditions, we observed no significant change in pH$_i$ (Supplementary Figure 7a, $n = 5$ experiments). A subsequent voltage step to −100 mV again elicited SpSLC9C1 activity. The normalized and background corrected resting activity under these conditions was $4.3 \times 10^{-3} ± 2.0 \times 10^{-2}$ (range $−2.5 \times 10^{-2}$ to $3.1 \times 10^{-2}$). Therefore, we believe that it is safe to estimate that the resting activity is below 3% of its maximum value (Supplementary Figure 7b).

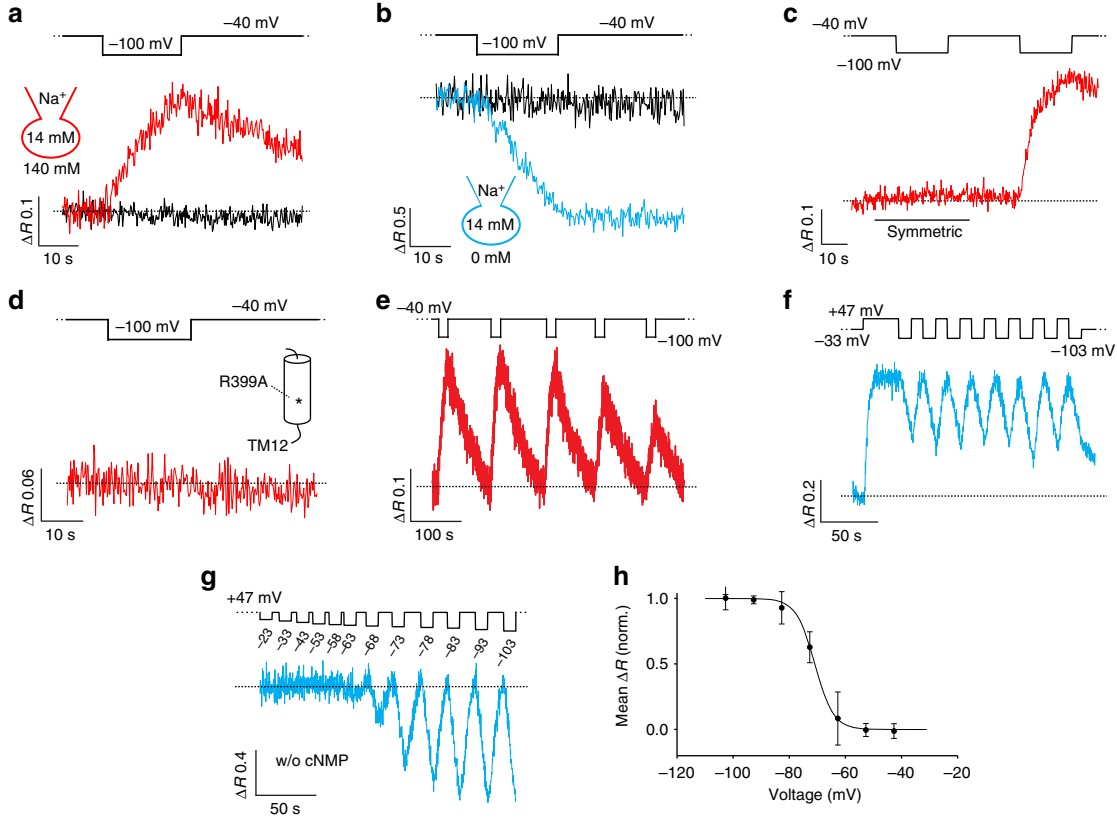

**Fig. 3** Voltage-clamp $pH_i$ fluorimetry of SpSLC9C1 activity. **a** Alkalinization induced by a 20 s step hyperpolarization to −100 mV using an inwardly directed $Na^+$ gradient ($[Na^+]_i$ 14 mM; $[Na^+]_o$ 140 mM; $pH_i = 7.2$; $pH_o = 7.4$, forward mode, red). No $pH_i$ change occurred in non-transfected CHO cells (black). **b** Acidification induced by a 20 s step hyperpolarization to −100 mV using an outwardly directed $Na^+$ gradient ($[Na^+]_i$ 14 mM; $[Na^+]_o$ 0 mM; $pH_i = 7.2$; $pH_o = 7.4$, reverse-mode, blue). **c** Perfusion with symmetric solutions (black line) abolished the net $Na^+/H^+$ exchange due to a lacking gradient ($pH_i = pH_o = 7.2$, $[Na]_i = [Na]_o = 14$ mM). Voltage induced net $Na^+/H^+$ exchange was restored when the cell was perfused with 140 mM $Na^+$ ($pH_o = 7.4$). **d** When Arg399 in T12 of the exchanger domain is replaced by Ala (R399A), $Na^+/H^+$ exchange was abolished. **e** $pH_i$ responses to repetitive voltage steps from −40 to −100 mV in a CHO-SpSLC9C1 cell. Dotted line indicates resting $pH_i$. ($[Na^+]_i$ 14 mM; $[Na^+]_o$ 140 mM; $pH_i = 7.2$; $pH_o = 7.4$). **f** Repetitive stimulation of SpSLC9C1 activity (reverse mode) in CHO cells that co-express the $H^+$-selective channel Hv1. Cells quickly recovered from acidification by activation of Hv1 at +47 mV. ($[Na^+]_i$ 14 mM; $[Na^+]_o$ 0 mM; $pH_i = 7.2$; $pH_o = 7.4$). **g** Voltage dependence of SpSLC9C1 activation was determined by stepping $V_m$ between −23 and −103 mV to +47 mV ($V_{1/2} = -70.4$ mV; $[Na^+]_i$ 14 mM; $[Na^+]_o$ 0 mM; $pH_i = 7.2$; $pH_o = 7.4$). **h** Normalized $\Delta R$ values were plotted against $V_m$ to yield the $V_{1/2}$ values by a fit with the Boltzmann equation ($V_{1/2} = -70.9 \pm 2.5$ mV, $s = 3.3 \pm 0.9$ mV, $n = 7$). Mean values are summarized in Table 1

## Cyclic AMP modulates gating currents and exchange activity.

Although the CNBD domain of SpSLC9C1 is suggestive, several ion channels and protein kinase A (PKA) orthologues that carry a CNBD domain are, in fact, not gated or activated by cyclic nucleotides[5,43,44]. Therefore, we examined the action of cyclic nucleotides on gating currents and $Na^+/H^+$ exchange (Figs. 2 and 4).

Cyclic AMP (1 mM) in the pipette shifted the $V_{1/2}$ of gating currents by 20 mV to −74.4 ± 6.4 mV, whereas cGMP (1 mM) was much less effective (Fig. 2b). The number of gating charges was similar in the absence and presence of cAMP or cGMP. Similarly, cAMP shifted the $V_{1/2}$ of gating-current activation of the VSD mutant R803Q by 21 mV to −96.8 ± 6.6 mV (Fig. 2c; Table 1). When cNMP binding was disabled by replacing a key Arg with a neutral residue in the phosphate-binding cassette of the CNBD (R1053Q)[45–47], charge movement in the absence of cAMP was not affected, but the cAMP-induced $V_{1/2}$ shift was abolished (Fig. 2d). Finally, this $V_{1/2}$ shift was not affected by the R399A mutation in the exchanger domain (Fig. 2e). This result demonstrates that binding of cAMP to the CNBD changes the VSD equilibrium.

Next, we examined whether cyclic nucleotides control exchange activity (Fig. 4). With cAMP (1 mM) in the pipette, the $V_{1/2}$ of exchange activity was shifted by 15 mV to −56.8 ± 2.7 mV (Fig. 4a, b; Table 1). Again, cGMP was much less effective (Fig. 4b). To compare $Na^+/H^+$ exchange without and with cAMP in the same cell, we rapidly photo-released cAMP from caged derivatives of cAMP (BCMCM-cAMP and BECMCM-cAMP)[48]. At −63 mV, reverse-mode activity produced only a small acidification. Photo-release of cAMP instantaneously stimulated exchange activity (Fig. 4c). The action of cAMP saturated after a few flashes (Fig. 4d). At higher light intensity, the number of flashes required to saturate the response was lower (Fig. 4d). The $V_{1/2}$ and $s$ of activation after photolysis (−55.3 ± 5.1 mV and 5.0 ± 0.6 mV ($n = 3$), respectively) was similar to that recorded in the presence of cAMP in the pipette (Fig. 4e). Finally, in the R1053Q mutant with disabled CNBD, the activation of $Na^+/H^+$ exchange by voltage was not altered, but the $V_{1/2}$ shift by cAMP was abolished (Fig. 4f). These results demonstrate that binding of cAMP to the CNBD modulates $Na^+/H^+$ exchange by shifting the voltage dependence of activation.

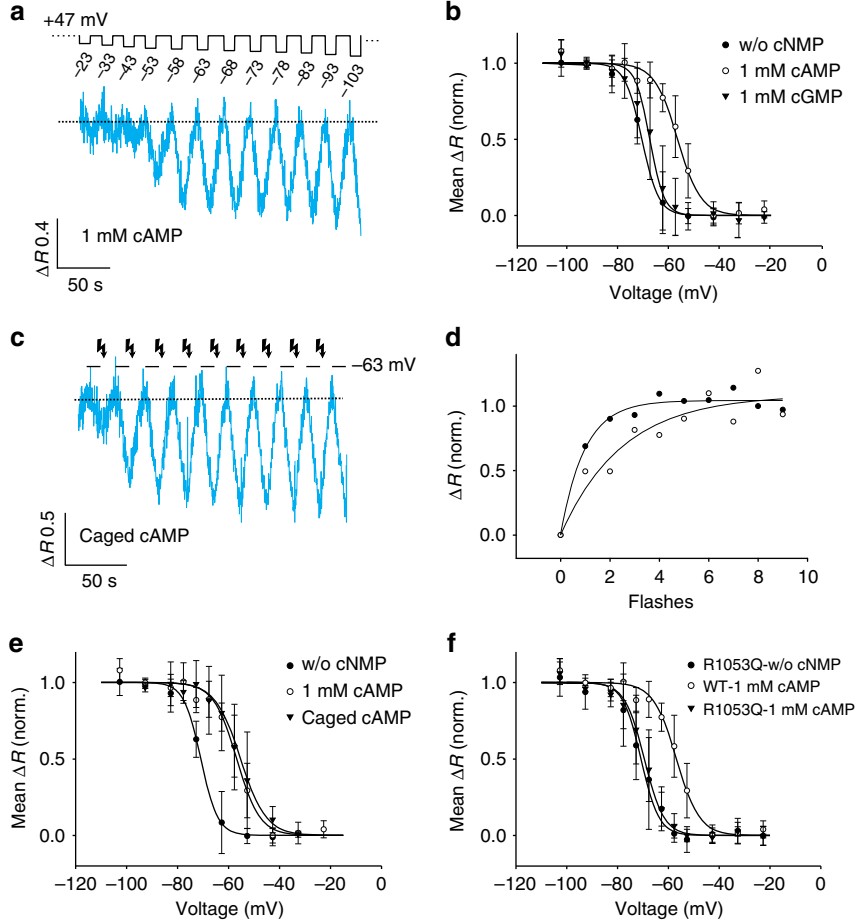

**Fig. 4** Modulation of $Sp$SLC9C1 activity by cAMP. **a** Action of 1 mM cAMP in the pipette solution on the voltage dependence of $Sp$SLC9C1 ($V_{1/2} = -53.4$ mV). ([Na$^+$]$_o$ 0 mM; [Na$^+$]$_i$ 14 mM; pH$_i$ = 7.2; pH$_o$ = 7.4). **b** Normalized $\Delta R$ values were plotted against $V_m$ to yield the $V_{1/2}$ values by a fit with the Boltzmann equation (w/o cNMP: $V_{1/2} = -70.9 \pm 2.5$ mV, $s = 3.3 \pm 0.9$ mV, $n = 7$; cAMP: $V_{1/2} = -56.8 \pm 2.7$ mV, $s = 4.6 \pm 1.5$ mV, $n = 7$; cGMP: $V_{1/2} = -67.8 \pm 5.4$ mV, $s = 2.8 \pm 1.0$ mV, $n = 9$). **c** Flash photolysis of caged cAMP enhanced $Sp$SLC9C1 activity at a holding voltage of $-63$ mV and saturated after 3–4 flashes, same conditions as in **a**. **d** Normalized $\Delta R$ values were plotted against the number of flashes. Black circles show values from **c**. Decreasing the light energy of the flash, saturation of $Sp$SLC9C1 activity required 5–6 flashes (white circles). **e** The shift of $V_{1/2}$ and $s$ values evoked by flash photolysis were similar to those using cAMP in the pipette (black triangles: $V_{1/2} = -55.3 \pm 5.1$ mV, $s = 5.0 \pm 0.6$ mV, $n = 3$; for comparison: w/o cNMP (black circles) and 1 mM cAMP (white circles). **f** Replacing the Arg1053 in the CNBD by Gln (R1053Q) maintained a wild-type-like $V_{1/2}$ in the absence of cNMP ($-70.8 \pm 4.8$ mV, $s = 3.6 \pm 1.1$ mV, $n = 5$), but strongly reduced the $V_{1/2}$ shift by cAMP ($-69.5 \pm 3.8$ mV, $s = 3.9 \pm 1.2$ mV, $n = 6$). Mean values are summarized in Table 1

**$Sp$SLC9C1 mediates the chemoattractant-induced alkalinization**. Stimulation of sperm from *S. purpuratus* and *Arbacia punctulata* with chemoattractant peptides evokes a transient hyperpolarization and a rapid alkalinization[9,14–18,49–51]. When the hyperpolarization was abolished, the alkalinization was abolished as well[17]. We studied whether this alkalinization is caused by Na$^+$/H$^+$ exchange via $Sp$SLC9C1. Using a rapid-mixing technique, we followed kinetically H$^+$ efflux and Na$^+$ influx in *S. purpuratus* sperm. Changes in pH$_i$ and intracellular Na$^+$ concentration ([Na$^+$]$_i$) were measured using BCECF and Asante Natrium Green-2, respectively. Stimulation with the chemoattractant speract elevated pH$_i$ and [Na$^+$]$_i$ with a similar dose dependence, time course, and latency (Fig. 5a, b). The relation between the latencies of Na$^+$ and pH$_i$ responses was linear (slope of 1) over two orders of speract concentrations (Fig. 5b inset), indicating that H$^+$ efflux and Na$^+$ influx are mechanistically coupled. To investigate whether $Sp$SLC9C1 mediates Na$^+$/H$^+$ exchange in sea urchin sperm, we studied the cAMP dependence of ion exchange by loading sperm with DEACM-caged cAMP[52]. We studied the action of cAMP pho-torelease on the resact-evoked pH$_i$ responses (Fig. 5c). For all

speract concentrations, the pH$_i$ response was faster and larger when cAMP was released (Fig. 5c). The exchange activity was analyzed by plotting the maximal slope of the pH$_i$ change in the absence (black) or presence (red) of cAMP uncaging (Fig. 5d). When cAMP was released, exchange activity was enhanced at all speract concentrations (Fig. 5d). As control, we studied speract-evoked voltage signals with and without uncaging cAMP. Voltage responses were not affected by uncaging cAMP (Fig. 5e), demonstrating that the enhanced exchanger activity is not due to a larger hyperpolarization.

Blockers of SLC9A exchangers, including amiloride, failed to inhibit $Sp$SLC9C1 in heterologous systems. Similarly, in sea urchin sperm, the speract-induced pH$_i$ signals were largely unaffected by amiloride (500 µM, Supplementary Figure 5e, f).

In western blots of flagellar membranes, a polyclonal antibody raised against $Sp$SLC9C1 recognized three proteins with apparent $M_w$ of 132, 137, and 146 kDa similar to the $M_w$ of 146.6 kDa predicted for $Sp$SLC91 and to the apparent $M_w$ of HA-tagged $Sp$SLC9C1 in CHO cells (Fig. 6a and Supplementary Figure 8). The two prominent protein bands at higher $M_w$ might represent dimers and tetramers. $Sp$SLC9C1 was less abundant in head

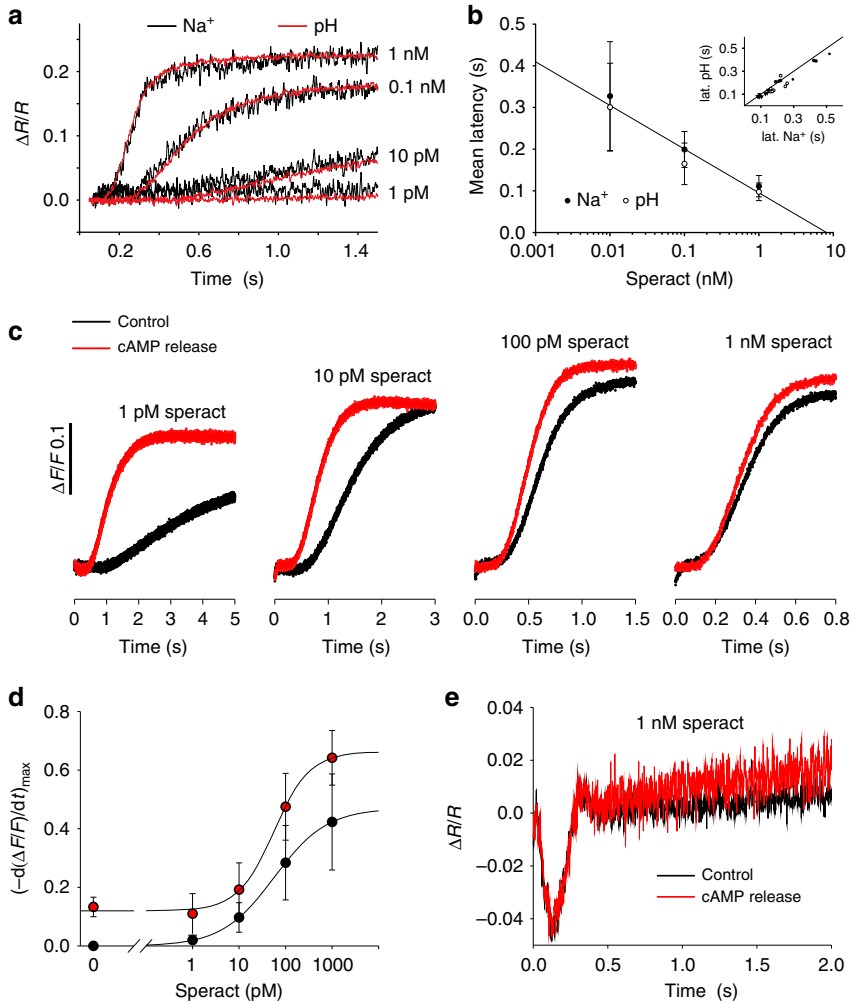

**Fig. 5** Sodium and proton fluxes in *S. pupuratus* sperm. **a** Speract-induced changes in fluorescence ratio ($\Delta R/R$) indicating $Na^+$ influx (ANG2, black) or proton efflux ($pH_i$) (BCECF, red); signals were scaled and superimposed. Speract concentrations are shown on the right. **b** Mean latency ± SD ($n = 7$) of $pH_i$ and $Na^+$ signal plotted vs. speract concentration. Inset: latency of $pH_i$ and $Na^+$ signal at different speract concentrations (10 pM black circles; 100 pM white circles; and 1 nM triangles). The line represents the identity function ($n = 7$). **c** Speract-induced alkalinization monitored by pHrodo Red fluorescence in the absence (black trace) or presence (red trace) of release of cAMP from DEACM-caged cAMP with a continuously pulsing UV-LED. UV light was applied during the entire recording time. **d** The maximal slope of the pHrodo Red time course was plotted vs. speract concentration in the absence (black) and presence (red) of cAMP released from DEACM-caged cAMP ($n = 3$). **e** Voltage recordings during speract stimulation (1 nM) in the absence (black) or presence (red) of cAMP released from DEACM-caged cAMP

compared to flagellar preparations (Fig. 6a and Supplementary Figure 8). Multiple bands may indicate heterogeneous posttranslational modification. Finally, in immunocytochemistry, the antibody stained the flagellum and also partially the head (Fig. 6b). These results confirm the presence of *Sp*SLC9C1 in *Strongylocentrotus purpuratus* sperm[53]. Altogether these experiments suggest that *Sp*SLC9C1 mediates the chemoattractant-induced alkalinization.

## Discussion

*Slc9* genes encode a large family of cation/proton-coupled antiporters, which fall into three subgroups: classical $Na^+/H^+$ exchangers of the SLC9A (also referred to as NHE) subtype, the SLC9B (NHA) subtype, and SLC9C (sNHE) subtype. The SLC9A subfamily has been studied extensively, whereas, apart from gene-inactivation studies, little is known about SLC9B and SLC9C (SLC9B1[54]; SLC9C1[2]). Here, 15 years after the discovery of the SLC9C1 gene, we demonstrate that *Sp*SLC9C1 represents an electroneutral $Na^+/H^+$ exchanger with a truly remarkable activation mechanism.

Like voltage-activated ion channels, *Sp*SLC9C1 is gated by movement of a voltage sensor; this movement is controlled by cAMP binding to a C-terminal cyclic nucleotide-binding domain. The VSD of *Sp*SLC9C1 harbors seven positively charged amino acids in the S4 motif. Assuming a simple two-state Boltzmann mechanism of VSD movement, we estimate for *Sp*SLC9C1 activation an effective valence or gating charge $g_q$ of 3.1 $e_0$, which is similar to that of Shaker $K^+$ channels ($g_q$ values of 2.5 $e_0$ per subunit[55] or 12–13 $e_0$ per tetrameric channel[56]). Thus, the number of charges transported across the membrane during gating is similar in *Shaker* channels and *Sp*SLC9C1. For the only other non-channel VSD in a lipid phosphatase of *Ciona intestinalis* sperm, $g_q$ values are 1–1.6 $e_0$[57,58].

How is *Sp*SLC9C1 activity controlled by the VSD? In general, solute carriers undergo cycles of conformational changes for upload and release of substrates on opposite sides of the membrane—a mechanism known as alternating access model[59] or rocking mechanism. A defining feature of this model is that ions or substrates themselves gate the conformational change and that solutes passively redistribute in response to extra- or intracellular

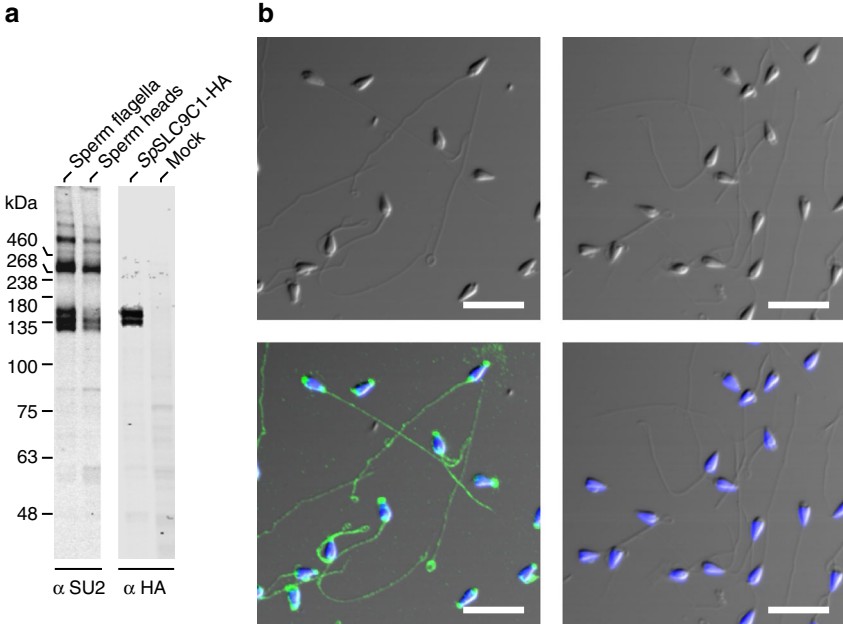

**Fig. 6** Analysis of *Sp*SLC9C1 expression in sea urchin sperm. **a** Representative western blot using protein lysates from flagella and heads of *S. purpuratus* sperm, from CHO cells heterologously expressing HA-tagged *Sp*SLC9C1 (*Sp*SLC9C1-HA), and from non-transfected CHO cells (mock). *Sp*SLC9C1 in flagella and heads was probed with the anti-*Sp*SLC9C1 antibody SU2 (left panel) and *Sp*SLC9C1-HA in CHO K1 cells was probed with an HA antibody (right panel, see also Supplementary Figure 8). **b** Immunocytochemical analysis of sperm stained with SU2 (left) and pre-immune serum control (right). Upper panels: bright-field (DIC) microscopy. Lower panels: overlay of DIC and fluorescence images (*Sp*SLC9C1, green; nucleus stained with DAPI, blue). Scale bars represent 10 μm

changes in substrate concentrations. By contrast, in *Sp*SLC9C1, movement of a channel-like VSD couples membrane voltage to ion exchange. We envisage two different gating mechanisms. In one model, at resting voltage, either ions cannot reach the binding site, because access is obstructed by a physical gate, or ions cannot be uploaded, because the binding affinity is low (non-accessible, Fig. 7a). Hyperpolarization opens the gate or enhances the binding affinity. Alternatively, at rest, ions can access and be uploaded to the binding site, but the rocking motion is blocked (non-rocking, Fig. 7a) and hyperpolarization enables this exchange motion. These mechanisms are summarized by an extended kinetic scheme of exchange activity[60] (Supplementary Figure 7c). A corollary of this scheme is that the exchanger—like ion channels—may have a small basal activity.

During their lifetime, sperm are exposed to vastly different aqueous conditions and compartments, involving large changes in external $pH_o$ and $[Na^+]_o$[61]. Gating of $Na^+/H^+$ exchange by voltage implies that extracellular changes are not relayed to the cytosol, unless triggered by a change in voltage that initiates rapid substrate redistribution. This mechanism might safeguard sperm from their environment and is functionally important for rapid periodic stimulation, while navigating in a chemical gradient. In conclusion, SLC9C1 represents a rapidly responding signaling molecule rather than a transporter for housekeeping $pH_i$ homeostasis.

This study also provides insight into the puzzling function of cAMP in sperm from marine invertebrates. Whereas the role of cGMP in chemotactic signaling has been established[62], the physiological function of cAMP remains elusive (Fig. 7b). With the regulation of *Sp*SLC9C1 activity by cAMP, previously unexplained observations fall into place. For example, the hyperpolarization-evoked alkalinization persists for several tens of seconds[9], whereas the hyperpolarization succumbs within a second[50]. The cAMP-induced $V_{1/2}$ shift of *Sp*SLC9C1 activation from $-72$ to $-55$ mV implies that $Na^+/H^+$ exchange now

also operates and maintains alkaline $pH_i$ near resting $V_m$ (about $-50$ mV). The alkalinization primes CatSper channels to open and promotes $Ca^{2+}$ entry[9]. Thus, cAMP, by maintaining alkaline $pH_i$, keeps CatSper channels primed. This mechanism enables CatSper to translate rapid periodic $V_m$ changes into periodic $Ca^{2+}$ signals while sperm swim on a periodic path in a chemical gradient[9].

How is cAMP synthesis controlled? A soluble adenylate cyclase (SACY) is the predominant source for cAMP in sperm[63,64]. There is evidence for a direct physical interaction between SLC9C1 and sAC[12,64]. What is the functional role of this interaction? Whereas in many species, activity of SACY is controlled by bicarbonate and $Ca^{2+}$[65–67], circumstantial evidence indicates that cAMP synthesis in sea urchin sperm might be directly or indirectly regulated by membrane voltage[49], or by $pH_i$[68], or some other mechanism. We speculate that *Sp*SLC9C1 confers voltage sensitivity to SACY either directly via its VSD or indirectly via alkalinization (Fig. 7b). In turn, cAMP regulates $Na^+/H^+$ exchange (Fig. 7b). Future studies need to address this potential reciprocal control of SLC9C1 and SACY in sperm of sea urchin, vertebrates, and mammals.

Functionally important amino-acid residues in the exchange domain are conserved between archaeal $Na^+/H^+$ exchangers, mammalian SLC9A members, and *Sp*SLC9C1 (Supplementary Figure 1). Moreover, the VSD and CNBD domains of *Sp*SLC9C1 are functional and feature all structural hallmarks of the respective domains in voltage-gated ion channels and cyclic nucleotide-sensitive proteins (Fig. 1 and Supplementary Figure 1). In comparison, mammalian SLC9C1 members display several variations of functionally important residues. For example, the ND motif, which is part of the cation-binding site and which is characteristic of 1:1 $Na^+/H^+$ exchangers, is replaced by a TS motif in mammalian SLC9C1 members. This might be a clue that mammalian SLC9C1 proteins acquired different functions. Finally, three of the four N-terminally located Arg or Lys residues in the S4 motif

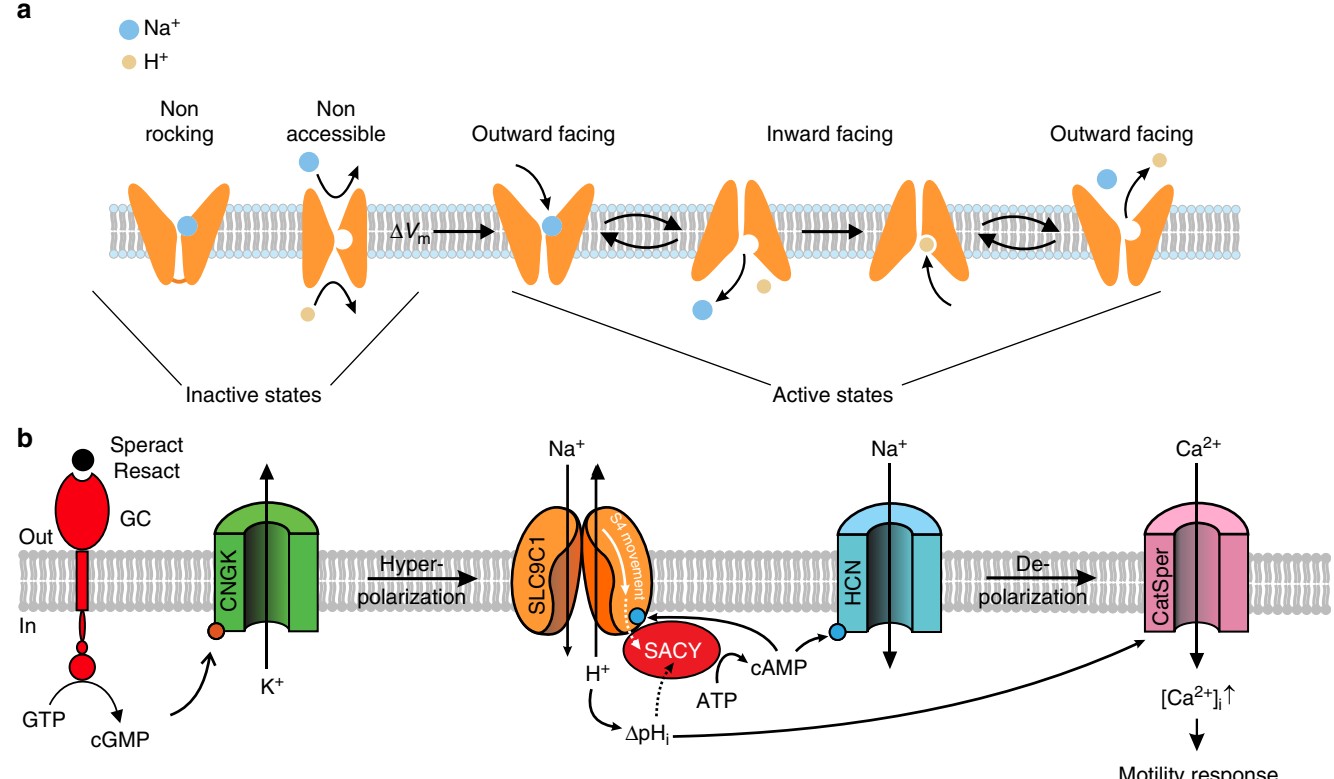

**Fig. 7** Models of SLC9C1 gating by voltage and cAMP. **a** Cartoon depicting that *Sp*SLC9C1 is active only when the voltage enables its activity. The VSD could gate exchange activity by providing access for ions to their binding sites either by removing a physical gate or by enhancing the binding affinity for uploading (non-accessible). Alternatively, in the resting state, ions have access to their binding sites, but the rocking mechanism is locked (non-rocking); voltage unlocks the rocking motion and thus allows switching between outward- and inward facing conformations. **b** Chemotactic signaling pathway in sea urchin sperm. Chemoattractant binding results in cGMP production and CNGK channel opening. The subsequent hyperpolarization activates SLC9C1. Alkalinization and cAMP production cooperate to open the CatSper channel. The intimate interaction between the exchanger and the soluble adenylate cyclase SACY is illustrated

of *Sp*SLC9C1 are missing in mouse and human orthologues (Fig. 1c), suggesting that voltage activation of mammalian SLC9C1 proteins may be different. Thus, our work provides a technical and conceptual blueprint for future studies of mammalian SLC9C1 orthologues.

## Methods

**Strongylocentrotus purpuratus sperm samples**. Collection of dry sperm was described previously[62]. In brief, 0.2–0.5 ml of 0.5 M KCl was injected into the sea urchin cavity to induce spawning. Spawned sperm (dry sperm) were collected using a Pasteur pipette and stored on ice.

**Measurement of $[Na^+]_i$, $pH_i$, and $V_m$ in S. purpuratus sperm**. We measured speract-induced changes in $[Na^+]_i$, $pH_i$, and $V_m$ by loading sperm samples with the corresponding dye (Asante Natrium Green 2/ANG2, TEFLabs, Austin, USA; BCECF-AM and pHrodo Red-AM, both from Molecular Probes, Eugene, USA; and di-4-AN(F)EP(F)PTEA[69] (kind gift of Dr. L. Loew) in a rapid mixing device (SFM-400, BioLogic, Claix, France)[62]. Dry sperm was suspended 1:6 (v/v) in Artificial Sea Water (ASW) containing (in mM); NaCl 423, CaCl$_2$ 9.27, KCl 9, MgCl$_2$ 22.94, MgSO$_4$ 25.5, EDTA 0.1, HEPES 10 at pH 7.8, and the respective dye (18 °C). Loading concentrations and times were: ANG2: 10 μM for 60 min (0.5% w/v Pluronic F127, Molecular Probes); BCECF-AM: 10 μM for 10 min; pHrodo Red-AM: 16 μM for 30 min (0.5% Pluronic F127); di-4-AN(F)EP(F)PTEA: 25 μM for 5 min (0.5% Pluronic F127). After 1:20 (v/v) dilution in ASW, sperm were allowed to equilibrate for 5 min. Subsequently, sperm were mixed 1:1 (v/v) with speract in ASW. ANG2 signals in sperm were recorded at a flow rate of 1.3 ml s$^{-1}$. BCECF and pHrodo Red signals were recorded at flow rates of 1.3 ml s$^{-1}$ (Fig. 5a) or 4 ml s$^{-1}$ (Fig. 5c). di-4-AN(F)EP(F)PTEA signals were recorded at a flow rate of 4 ml s$^{-1}$. Fluorescence was excited by pulsed LED light (SpectraX Light Engine, Lumencor, Beaverton, USA or M490L3, Thorlabs, Newton, USA) with a frequency of 10 kHz or by high power LEDs (Thorlabs) with frequencies up to 80 kHz. Emission was recorded by photomultiplier modules (H9656-20 and C7169, Hamamatsu Photonics, Japan). The signal was amplified and filtered by a lock-in

amplifier (7230 DualPhase, Ametek, Paoli, USA). Data acquisition was performed with a data acquisition pad (PCI-6221, National Instruments, Austin, USA) and Biokine Software v.4.49 (BioLogic). For $[Na^+]_i$ recordings, excitation light passed a 485/25 nm filter (AHF, Tübingen, Germany). The emitted fluorescence passed a 536/40 nm filter (Semrock, Rochester, USA). For BCECF recordings, excitation light passed a 452/45 nm filter (Semrock). BCECF fluorescence was recorded in dual-emission mode using Brightline 494/20 nm and 540/15 nm filters (Semrock). The $pH_i$ signals represent the ratio of $F_{494}/F_{540}$. They are the average of at least two recordings and are depicted as the percent change in ratio ($\Delta R/R$) with respect to the first 10–20 data points after mixing. For pHrodo Red recordings, excitation and emission were at 572/15 and 628/40 nm, respectively. For $V_m$ recordings, excitation light passed a 513/18 nm filter (Semrock); di-4-AN(F)EP(F)PTEA fluorescence was recorded in dual emission mode using Brightline 549/15 and 705/72 nm filters (Semrock). The $V_m$ signals (ratio $F_{549}/F_{705}$ ($R$); average of at least four recordings) are depicted as the percent change in ratio ($\Delta R/R$) with respect to the first 10 data points. The baseline control (ASW) was subtracted from the speract-induced signals. To manipulate cAMP in sperm during chemoattractant stimulation, sperm were incubated with 30 μM DEACM-caged cAMP for 30–60 min. Uncaging was performed in the stopped-flow cuvette by a 360 nm LED (Thorlabs) with 58 kHz.

**Generation of a polyclonal anti-SpSLC9C1 antibody**. Peptides comprising amino acids (aa) 574–591 (EFADMMEEARLRMLKAEK), aa 857–871 (MVDNKKIL-RELKHIS), aa 937–957 (KMKRLMNAPSSIPPPPPENLL), and aa 1111–1126 (GWTQEKVKLHLERGYL) were synthesized and coupled to BSA via a cysteine that was introduced at the N terminus of peptides. Rabbit antibodies directed against a mixture of these peptides, resulted in two polyclonal antibodies SU1 and SU2. The SpSLC9C1 antibody was purified from antisera by affinity purification using the four peptides. All steps for antibody production were performed by Davids Biotechnology, Regensburg, Germany.

**Preparation of heads and flagella from S. purpuratus**. Sperm flagella and heads were separated as described[9] with one modification: instead of shearing with a 24-G needle, the sperm suspension was sheared 20 times by centrifugation for 30 s at 75 × g and 4 °C through the net of a 40-μm cell strainer (BD Biosciences).

**Cell lines**. CHO-K1 cells were obtained from European Collection of Authenticated Cell Cultures (ECACC, catalog number 85051005). Mycoplasma tests were carried out regularly, once a year. The results of the mycoplasma tests are available upon request.

**Western blotting**. Sperm flagella and heads from *S. purpuratus*, CHO K1 cells stably expressing HA-tagged *Sp*SLC9C1 (*Sp*SLC9C1-HA), and CHO K1 control cells (mock) were lysed by sonication in a hypotonic buffer containing 10 mM Hepes/NaOH, pH 7.4, 2 mM EDTA, and protease inhibitor mixture mPIC (Sigma Aldrich, St. Louis, USA). The suspension was centrifuged for 10 min at $500 \times g$. The supernatant was used for Western blotting. Twenty µg protein per lane were separated by SDS-PAGE using a precast NuPage Novex 7% Tris-Acetat Protein Gel (Thermo Fisher Scientific, Waltham, USA). The samples were heated for 5 min at 95 °C prior to separation. Protein Marker VI (AppliChem, Darmstadt, Germany) and HiMark Pre-Stained Protein Standard (Thermo Fisher Scientific) were used as molecular weight markers. Proteins were transferred onto an Immobilon FL PVDF membrane (Merck Millipore, Darmstadt, Germany), probed with antibodies, and analyzed using the Odyssey Imaging System (LI-COR, Bad Homburg, Germany). All figure panels were taken from the same western blot. Figures were prepared using CorelDrawX6 and Photo-PaintX6 software (both from Corel Corporation). Primary antibodies were: *Sp*SLC9C1-SU2 (1:3000), rat-anti-HA (1:3000; Roche Applied Science catalog no. 11867431001, Penzberg, Germany). Secondary antibodies were as follows: IRDye680 and IR800 antibodies (LI-COR, 1:25,000).

**Cloning of *S0p*SLC9C1**. Four sets of primer pairs, designed on the annotated sequence for the *Strongylocentrotus purpuratus* sperm-specific sodium proton exchanger, NM_001098457, were used to obtain the full-length sequence by PCR amplification on a *Strongylocentrotus purpuratus* testis library. Primers C0600, C0601, C0602, and C0603 were used to amplify the 5′ part of the sequence (bp 1 to 1444) by to rounds of PCR. Primer C0600 introduces a BamHI followed by a perfect Kozak sequence[70] preceding the start codon. Primers C0601 and C0602 destroy the internal BamHI site at position 624 by a silent mutation. Primer C0603 introduces an EcoRI site. Primers C0604 and C0605 were used to amplify bases 1432 to 2734. Primer C0604 introduces a EcoRI site, primer C0605 introduces a XhoI site by silent mutation. The 3′ part (bp 2723 to 3975) was amplified with primers C0610, C0606, C0607 and C0608 in two rounds of PCR. Primers C0606 and C0607 destroy the internal XhoI site at position 3361 by a silent mutation. Primer C0610 introduces a XhoI site. Primer C0608 adds a sequence for an HA-tag to the 3′ end, followed by a stop codon and an XbaI site. The primer sequences were:

ACAGGATCCACCATGAAGAAGAGAGTCGTGAAATTG (C0600),
CGACGGGATCTGTCGCACTCATG (C0601),
CATGAGTGCGACAGATCCCGTCG (C0602),
CCAGAATTCTCAGTAGAGTCTGGATGG (C0603),
CTGAGAATTCTGGGCATGAGTGACATC (C0604),
TCTCTCGAGAGTGGTTGAGAATGGTG (C0605),
ACTCTCGAGAGACTATCCATGAACTCC (C0610)
GCCACGCTCAAGATGTAACTTCAC (C0606),
GTGAAGTTACATCTTGAGCGTGGC (C0607), and
TCTTCTAGATTAGGCGTAGTCGGGCACGTCGTAGGGGTAAACGTT GACCCTAGGGGGCC (C0608).

All three resulting PCR fragments were cloned together into vector pcDNA3.1 (+) (Invitrogen, Carlsbad, USA) to obtain the full-length clone.

**Generation of stable CHO cell lines of *Sp*SLC9C1 and *h*Hv1**. CHO K1 cells were electroporated with pc3 sNHE-HA or with pc3 *h*Hv1 using the Neon 100 Kit (Invitrogen, Carlsbad, USA) and a MicroPorator (Digital Bio) according to the manufacturer's protocol ($3 \times 1650$ mV pulses with a 10-ms pulse width). Cells were transferred into complete medium composed of F12 plus GlutaMAX (Invitrogen) and 10% fetal bovine serum (Biochrom, Berlin, Germany). To select monoclonal cells stably expressing *Sp*SLC9C1 or *h*Hv1, the antibiotic G418 (1200 mg ml$^{-1}$; Invitrogen) was added to the cell culture medium 24 h after the electroporation. Monoclonal cell lines were identified by immunocytochemistry using a rat-anti-HA antibody (Roche Applied Science) or by electrophysiological recordings.

**Immunocytochemistry**. Sperm samples were stained as previously described[9]. Antibodies were diluted 1:400 (*Sp*SLC9C1-SU1) and 1:750 (*Sp*SLC9C1-SU2) in 0.1 M phosphate buffer (pH 7.4) in the presence of 0.5% Triton X-100 and 5% chemiblocker (Millipore). Primary antibodies were incubated overnight at 4 °C and visualized by goat-anti-rabbit A488 (1:500, 20 min incubation at RT, ThermoFisher Scientific A-11034, Rockford, USA). In CHO cells, *Sp*SLC9C1-HA was probed by a rat-anti-HA antibody (1:1000, 1 h incubation at RT, Roche Applied Science) and visualized by goat-anti-rat A488 (1:400, 20 min incubation at RT, ThermoFisher Scientific A-11006). Membrane sheet preparations were obtained by sonication (0.1 s, 5%, Vibracell, Sonics & Materials, Newtown, USA) of CHO cells plated on poly-lysine (0.1 mg ml$^{-1}$, Sigma Aldrich) coated glass coverslips (Marienfeld-Superior, Lauda-Königshofen, Germany) in HEPES-buffered solution (1 mM DTT, 0.2% mPIC v/v). As marker for outer plasma membrane, CHO cells were

additionally transfected with membrane-bound CAAX-RFP protein. For labeling the ER, a calnexin antibody was used (mouse-anti-calnexin, 1:500, Abcam ab31290, Cambridge, UK).

**Solutions for patch-clamp recordings**. Intracellular Solution (IS, in mM): NaCl 10, KAsp 130, EGTA 10, MgCl$_2$ 1, Na$_2$ATP 2, HEPES 1, adjusted to pH 7.2 with KOH (3 M). Extracellular Solution (ES, in mM): NaCl 140, KCl 5.4, MgCl$_2$ 1, CaCl$_2$ 1.8, HEPES 5, Glucose 10, adjusted to pH 7.4 with NaOH (1 M). ES-NMDG, NaCl substituted for *N*-methyl-D-glucamine (NMDG) and was adjusted to pH 7.4 with HCl.

**Single-cell fluorimetry**. We recorded in the whole-cell configuration changes in pH$_i$ from CHO cells expressing *Sp*SLC91 (wt), either stably expressed or using transient expression, or the respective *Sp*SLC91 mutants, using transient expression. For the determination of the voltage dependence of transport activity, we co-expressed *h*Hv1 with *Sp*SLC91 (wt), or the mutants *Sp*SLC9C1-R1053Q or *Sp*SLC9C1-R803Q. Cells were loaded with BCECF (10 µM) via the pipette and were excited with a Photon Technology International DeltaRam X$^{TM}$ monochromator (PTI, New Jersey, USA). BCECF fluorescence was recorded in dual-excitation mode at $440 \pm 6.25$ nm and at $480 \pm 6.25$ nm with 5-Hz frequency (100 ms nm$^{-1}$). Emitted light passed a dichroic mirror (500 nm LP) and a 525/15 nm filter (Semrock) and was detected by a photomultiplier system (Model 814, PTI). The pH$_i$ signals represent the ratio of $F_{480}/F_{440}$. We used a gravity-driven perfusion system. Temperature of ES and ES-NMDG solutions was set to 28 °C by a HPT-2 Heated Perfusion Tube (ALA Scientific Instruments Inc. St. Farmingdale, USA). For analysis of $V_{1/2}$ and slope (s) of *Sp*SLC9C1 activity, the initial slope of the $\Delta R$ signal was fitted by linear regression. The resulting slopes were plotted against voltage and were fitted to a Boltzmann equation $\Delta R(V) = \Delta R_{max} / \left(1 + \exp(V - V_{1/2})/s\right)$ with $s = (kT)/(q_g \times q_e)$. $k = 1.38 \times 10^{-23}$ J K$^{-1}$, $T = 301.15$ K (28 °C), and $q_e = 1.6 \times 10^{-19}$ As. $V_{1/2}$ is the potential where $\Delta R$ $(V) = \Delta R_{max}/2$. To determine mean $\Delta R$ values across data sets, we normalized each dataset to the corresponding parameters of the Boltzmann fit. For experiments with caged compounds, we loaded cells with membrane-permeant caged cyclic BECMCM-caged cAMP (10 µM) for 30 min prior to measurement and included also 100 µM BCMCM-caged cAMP in the pipette solution. Flash photolysis was achieved with short UV pulses (~1 ms) via a Xenon flash lamp System (JML-C2, Rapp OptoElectronic, Hamburg, Germany); light was passed through a UV filter (UV-2 250-375, Rapp OptoElectronic). Light energy was adjusted through the loading voltage of the lamp's capacitor and by neutral density gray filters (300 V, OD1 in Supplementary Figure [3]a; 200 V, OD0.4 in Supplementary Figure [3]b).

The calibration procedure for BCECF fluorescence to yield pH$_i$ by the pseudo-null-point method[33] was described previously[9]. Briefly, wild-type or *Sp*SLC9C1 expressing CHO cells seated on glass coverslips were loaded with 10 µM BCECF-AM for 10 min. Coverslips were placed into a home-built perfusion chamber on an Olympus cell$^R$ single-cell imaging system. Fluorescence at 540 nm was recorded ratiometrically by alternating excitation using 430/20 nm and 470–490 nm filters, yielding $R = F480/F430$. The pH$_i$ pseudo-null-point solutions contained defined concentrations of weak acid (propionic acid) and weak base (ammonium chloride) in ES solution. In the neutral form, weak acids and bases permeate the plasma membrane resulting in a transient change in pH$_i$. The extent and direction of the change in pH$_i$ can be predicted by pH$_{null}$ = pH$_o$ $- 0.5 \log[A]/[B]$, wherein $[A]$ refers to an acid and $[B]$ to a base. The concentrations of propionic acid (5 mM) and ammonium chloride (0.05, 0.32, 1.99, 12.56, 79.25 mM) yielded pH$_{null}$ solutions of 6.4, 6.8, 7.2, 7.6, and 8.0, respectively. To prevent side effects due to osmolarity issues, the NaCl concentration of the ES solution was respectively adjusted (134.95, 134.68, 133.01, 123.44, 55.75 mM for pH$_{null}$ of 6.4, 6.8, 7.2, 7.6, and 8.0).

**Gating currents**. We recorded gating currents in CHO cells expressing *Sp*SLC9C1 (wt), *Sp*SLC9C1-R1053Q, SpSLC9C1-R399A, or *Sp*SLC9C1-R803Q in ES at 28 °C. Online P/N leak subtraction was performed with four pre-pulses (P/4) opposite to pulse polarity to subtract linear currents due to leakage or capacitive artifacts (Clampex V1.10.2.0.12, MDS Analytical Technologies). Voltage steps ranging from +15 to −155 mV in steps of 10 mV were applied. Off-gating currents were integrated over time to yield net charges ($Q$). To quantify the voltage dependence of charge movement, we fitted $Q/V$ curves to a Boltzmann equation, defined as $Q(V) = 1/\left(1 + \exp\left(V - V_{1/2}\right)/s\right)$ with $s = (kT)/(q_g \times q_e)$. Mean gating charges ($Q/Q_{max}$) were determined by normalizing each data set to the corresponding parameters of the Boltzmann fit. For the *Sp*SLC9C1-R399A mutant, not all gating - current recordings were performed with the same voltage protocol. The number of charges ($q_g$) involved in the gating process was determined from $q_g = (kT)/(s \times q_e)$.

**Data availability**. The data that support the findings of this study are available from the corresponding authors upon request.

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

## Acknowledgements

We thank Heike Krause for preparing the manuscript and Dr. René Pascal for artwork. We thank Dr. Leslie Loew (University of Connecticut Health Center) for providing a sample of Di-4-AN(F)EP(F)PTEA. The data reported in the paper are presented in the main text and in the supplementary materials.

## Author contributions

U.B.K. and R.S. designed the project; F.W. and R.S. performed patch-clamp fluorimetry and stopped-flow experiments; W.B. cloned SpSLC9C1 and all mutants; H.G.K. designed antibodies and performed western blotting; E.G. performed electrophysiological experiments. F.W. performed immuno-cytochemistry; T.S. performed stopped-flow experiments; U.B.K., R.S., and F.W. wrote the manuscript; all authors edited the manuscript.

## Additional information

**Competing interests:** The authors declare no competing interests.

