## [Peer Review File · Nature Communications]

Reviewers' comments:

Reviewer #1 (Remarks to the Author):

General Comments: This paper addresses the issue of the properties and role of the SLC9C1 protein in sea urchin sperm, although the results also have general importance to the role of SLC9C1 in mammalian sperm. Although it has been assumed that SLC9C1 plays some role in solute transport, perhaps Na/H exchange, no functional studies have been successful, despite the fact that genetic KO of mouse SLC9C1 results in immotile sperm and infertile males. Here using a variety of experimental approaches, the authors convincingly demonstrate that heterologously expressed spSLC9C1 is an electroneutral Na/H exchanger that is regulated by two additional appended regulatory domains, a voltage-sensing domain (VSD) enabling changes in Na/H exchange in response to membrane potential and a cyclic nucleotide binding domain (CNBD) allowing regulation of exchanger activity by increases in cytosolic cyclic nucleotides. The results show clearly that both voltage and CNs shift the range of activation of Na/H exchange. Furthermore, the authors establish that regulation of alkalization in intact sea urchin sperm exhibit features consistent with the properties of the cloned spSLC9C1.

The paper addresses a long-standing, important question in sperm physiology and is generally clearly written. It is likely to many future studies that further probe the physiological role of SLC9C1's in sperm function, and also more mechanistic studies of how VSDs and CNBDs may regulate exchanger function.

One point. The authors seem to imply that the spSLC9C1 exchanger has no activity unless activated by voltage or with cyclic nucleotides. I'm not sure there is any data that allows this inference. There may be some low, but finite rate of exchange which is just enhanced by whatever it is the VSD (and CNBD) does. This is a minor point, but perhaps of value in how one approaches future mechanistic studies and the issue is raised in regards to some of the figures below. This issue also arises in regards to the cartoon of Figure 7A, in terms of whether the exchanger can ever be considered to be fully inactive. For many allosteric proteins, including channels, there is a finite active probability even in the fully resting condition, with that probability influenced by the VSD equilibrium or ligand binding equilibrium.

I have no serious concerns, but below I raise many points that might benefit from additional clarification or elaboration.

Specific Comments:

1. Abstract: "cAMP directly modulates the voltage dependence of gating" . Although the meaning of the authors is clear, more exactly it seems that cAMP is shifting the range of activation by voltage.
 2. Abstract wording of last line. "endows the ... exchanger in sperm to produce a rapid" would be better as "endows the ... exchange with the ability to produce a rapid"
 3. p. 3 middle "mouse SLC9C1 is non-functional in heterologous systems" . Were attempts to measure gating currents done? I gather that the pH fluorimetry experiments should have shown something similar to what is reported in the present manuscript, if mouse SLC9C1 were functional.
 4. p. 3 ... suggest "which serves as a switch to activate"
 5. Last sentence of Intro suggestion: "our results now enable future study of the commonalities and"
 6. p. 4. Title of first section. Comparison of putative protein topology isn't really structure, is it?
 7. Figure 2. The gating currents are much larger in the +cAMP example. Is that simply cell variability or a consistent finding?
 8. Figure 3. Although panels A and B show the Na concentrations in the experiment, I think it would also be useful to have the specific cytosolic and extracellular pH listed in the legends, as is done for Fig. 3C. This would also be helpful for Figure 4 and the supplementary figures, where appropriate.
- Figure 3A vs. 3B. Why does the alkalization begin to fall after return to -40 mV in A, but the acidification persists in B? Is there some other pathway by which protons can enter the COS cells in A or might this be basal NHE activity? In both A and B, at -40 mV it is presumed there is no

basal Na/H exchange at -40 mV, but that really isn't known, is it? In B, there is no extracellular Na, so NHE counted remove protons from the cytosol anyhow. Might the slow loss of alkalization in A reflect some very low rate of basal NHE activity even at -40 mV. Is the much larger size of the acidification signal in B (relative to alkalization in A) significant? Is there some sidedness to the net flux rate?

Figure 3C. I just want to suggest that this important panel does NOT show that exchange is not occurring, but that there is no NET proton flux.

Figure 3F-G-H, Figure 4A,C. The duration of each hyperpolarization leading to an acidification signal used to measure effect seems too short to reach a steady-state. Thus, the mean ΔR vs. voltage plots may be skewed by the fact that true steady-state is not measured. When comparing to cAMP this is not a problem, if the rates of NHE activation are similar, but that might not be the case. I don't think this is a serious problem, but it does seem curious that measurements were apparently made prior to steady-state.

On this topic, Figure S3C indicates that once an Hv1 expressing COS cell is made alkaline, at least over the time course of these experiments, there is no mechanism to low pH in the cells. Some comment somewhere about the intrinsic voltage-dependence of Hv1 might be helpful to readers. My understanding is that Hv begins to activate around -50 mV. I think the authors want to use Fig. S3C to infer that changes in pH signals in cells coexpressing Hv1 and spSLC during steps between -23 and -113 mV won't be impacted by Hv1. But I'm not sure that is fully justified. At those potentials (-33 to -58 mV, Fig 3G) where no response is seen in the absence of cNMP, might the ongoing activation of Hv1 over this range mute any ability to see an NHE effect. Is there a relatively specific inhibitor of Hv1 that could be used to test whether Hv1 activity might be compromising signals at more positive voltages? Overall, I don't see how any issues with such details impacts on the basic conclusions regarding effects of cAMP, but it just seems that some details of the methodologies could benefit from more description.

9. figure 4C. It might be useful to double the vertical scale on this panel, so the amplitude of the signals are more like in panel A.

10. top of page 8. In saying that coexpression with Hv1 overcomes these limitation, it might be useful to readers to explain why that is.

11. Regarding the explanation of Figure S3C in the text, it is stated that stepping to negative values from +47 mV did not change pH. I think this may leave an incorrect impression. I presume that pH did not change, since there was mechanism available to reacidify the cells. And the absence of a pH change may be taken to indicate that Hv1 will not influence pH signals over those more negative voltages. However, Hv1 gating is altered in this range, so if there is a simultaneous process altering pH, then Hv1 gating in that range may impact on the pH signals, making measurements of spSLC9C1 activity not as quantitative as one might like.

12. bottom of p. 9. "CNBD modulates VSD movement" This may or may not be correct. I think what the data reveal are that the CNBD influences the VSD equilibrium. That probably has nothing to do with movement per se.

13. Fig. S6C. It would be useful to indicate that the data with amiloride are without cAMP.

14. 2nd paragraph on p. 12. This touches on an issue already raised above. The question is, is activation of the VSD obligatory for any exchange to occur and the authors seem to assume the answer is affirmative. Physiological, that assumption may be largely correct, but mechanistically it will be an interesting question to be addressed in the future.

15. p. 13. "During their lifetime sperm are exposed" particular sperm or all sperm? 'unless triggered by a voltage pulse" Do sperm experience rapid voltage pulses?

16. near bottom. "This mechanism enables CatSper translating rapid..." change to "enables CatSper to translate rapid"

Reviewer #2 (Remarks to the Author):

The paper "The solute carrier SLC9C1 is a Na⁺/H⁺ exchanger gated by an S4-type voltage

sensor

and cyclic nucleotide binding" F. Windler and colleagues describe the characterization of SLC9C1 as an exchanger modulated by voltage and cAMP.

Alkalization is a key parameter to sperm physiology. Despite its importance the key molecule regulating pH in sperm from different species is not established. The sperm-specific Na⁺/H⁺ exchanger has been proposed to be regulated by voltage and cyclic nucleotides due to its amino acid sequence. However, this has not been proven experimentally. This manuscript is a nice addition to the deciphering of a long standing proven question about the gating of the sperm specific Na⁺/H⁺ exchanger.

In general, the paper is well written and easy to follow. The experiments are good and ingeniously planned with appropriate controls for most of the experiments. My recommendation is that this work should be accepted for publication.

However, I do have some comments that should be addressed/clarified before publication:

1. For Western blotting and immunofluorescence experiments (Fig. 6), the use of knock-out animal is the best control. This is almost impossible with sea urchin spermatozoa. At least, some control experiments to verify the antibody specificity should be performed with the primary antibody in the presence of excess amount of the antigen peptide. Signals from sperm head could be non-specific and control experiment with antigen peptide could resolve this issue. The authors claimed that multiple band observed in Western blots may be due to posttranslational modifications. Is there previous evidence of such modification for other Na⁺/H⁺ exchangers? Do these modifications may alter their function?

2. Endogenous Na⁺/H⁺ exchangers could be observed with this technique under different experimental and stimulating conditions?

3. Figures 3E and 3F show that the Run Down of SpSLC9C1 depends on the test pulse (or ionic compositions of the experimental media). The authors should comment about an explanation for this phenomenon?

4. From the text it is not quite clear if SpSLC9C1 and hHv1 were co-expressed in all the experiments with CHO K1 cells expressing SpSLC9C1 in patch-clamp fluorometry. It is indicated in Methods that stable cell lines (CHO K1) were produced by electroporation of pc3 sNHE-HA or pc3 hHv1 (but not the combination). This should be clarified in Methods

5. In general, absolute R values of BCECF (F480/F440) are more informative to compare absolute pHi values (at least the difference between cells non-expressing and expressing SpSLC9C1). However, only delta R values instead of R values were used in all figures including Fig. S3. The reason to use delta R instead of R should be mentioned in the text.

6. Authors utilized the null point method to calibrate pH. These experiments indicated that pHi value and potency of cytoplasmic pH buffer in the resting condition was not altered by the expression of SpSLC9C1. This is a quite important conclusion. However, the experimental procedure is not described in detail and the cited references are not sufficient to understand the procedure. Detailed experimental procedure should be described in the text.

7. Immunofluorescence signal of SpSLC9C1 in HEK K1 cells shows non-homogenous distribution (forming some patches) in the plasma membrane (Fig S2). The gating current indicates that SpSLC9C1 was really expressed in the plasma membrane without question, but some comments would be required for this unusual expression pattern.

8. SpSLC9C1 has an additional TM in the N-terminus compared to mammalian SLC9C1 (Fig. S1). If the authors have some insights about this difference related to the successful expression, it should be mentioned in the text.

9. To establish the specificity of the CNBD the authors should do a concentration study and not use a practically saturating concentration (1 mM) of the compounds. The endogenous cAMP is significantly lower under physiological conditions so the authors should discuss if the exchanger is regulated at lower concentrations.

10. Not being an expert in NHE, I consulted some of the given references. I noticed that references 14 and 18 (from the authors), are cited to highlight the importance of pH in chemoattraction, but actually these papers were reported to minimize the role of pHi changes in sea urchin sperm physiology. For example, the abstract of reference 18 states "We conclude that

an elevation of pHi is required neither to open Ca²⁺-permeable channels nor to control the chemotactic behavior". Another example is in the discussion section, with the sentence: "The alkalization primes CatSper channels to open and promotes Ca²⁺ entry", the work cited by the authors (9) demonstrates the presence of CatSper in sea urchin sperm but it does not demonstrate its fundamental participation in chemotaxis as huge concentrations of nonspecific blockers were used to inhibit chemotaxis. The authors should at least cite other evidence supporting the key role of CatSper in chemotaxis. This imprecise use of references is misleading and contributes to a distortion in the way readers perceive the development of the field and an unfair attitude in Biological Sciences, it should be corrected.

Minor point:

1. In page 11 line 4, "resact-evoked" should be "speract-evoked"

Reviewer #3 (Remarks to the Author):

This paper by Windler et al., report a characterization of an orphan sperm-specific family of solute carrier (SLC9C1) that features a unique structure: An exchanger domain, a VSD, and a CNBD. Since 2003 SLC9C1 has been suspected to serve as Na⁺/H⁺ exchanger that controls intracellular pH (pHi) in mammalian sperm and that might be operated by voltage. SLC9C1 gene knockout renders sperm immotile and male mice infertile, however, the pHi of SLC9C1^{-/-} sperm is not altered, suggesting probably of a compensation mechanism,, study published in 2007. These two pioneer studies have suggested that SLC9C1 may operate as Na⁺/H⁺ with an important physiological role.

The major claim of the paper that SLC9C1, Na⁺/H⁺ exchanger is also operated by voltage is somehow predictable. For someone very familiar with ion channels can predict a large aspect of the study. The cyclic nucleotide binding domain (CNBD) a structure that binds cAMP and cGMP is known for years as an important regulator for several ion channels. The structure and the operation mode of the voltage sensor domain (VSD) in ion channels are much known. The authors used mutations that are known to either affect CNBD of the VSD domain.

Although the data are of interest to others in the community, the finding is not novel per se. Wang, D., et al., King, S.M., Quill, T.A., Doolittle, L.K. & Garbers, D.L. A new spermspecific Na⁺/H⁺ exchanger required for sperm motility and fertility. *Nat. Cell Biol.* 5, 1117-1122 (2003). Wang, D. et al. A sperm-specific Na⁺/H⁺ exchanger (sNHE) is critical for expression and in vivo bicarbonate regulation of the soluble adenylyl cyclase (sAC). *Proc. Natl.Acad. Sci. USA* 104, 9325-9330 (2007).

I should give credit though to the authors for this interesting and nice characterisation that was somehow expected to come out.

I have few suggestions for a full characterization.

1- The V_{1/2} value of the off-gating current is very shifted to negative voltages! This means that at a resting membrane potential of -50mV of a sperm the voltage sensor is permanently in ON position. Question, what is the resting membrane potential of an Urchin Sperm? This is important, to have an idea on the state of the VSD and therefore the Na⁺/H⁺ activity at rest. Another point what ion channel is responsible of changing voltage in sperm? Can this or these channels be inhibited? This important to show the physiological role on the SLC9A.

2- Delete the VSD and CNBD on SLC9C1 and study the Na⁺/H⁺ properties and vice versa attach a VSD or/and CNBD to a mammalian Na⁺/H⁺ and study if this regulated by these added structures.

Response to the referees:

We thank the referees for their time to evaluate the manuscript, their overall positive assessment of the work, and the constructive criticism. Following the referees' suggestions, we performed additional experiments and revised the manuscript.

Response to referee #1:

General Comment:

The authors seem to imply that the spSLC9C1 exchanger has no activity unless activated by voltage or with cyclic nucleotides. I'm not sure there is any data that allows this inference. There may be some low, but finite rate of exchange which is just enhanced by whatever it is the VSD (and CNBD) does. This is a minor point, but perhaps of value in how one approaches future mechanistic studies and the issue is raised in regards to some of the figures below. This issue also arises in regards to the cartoon of Figure 7A, in terms of whether the exchanger can ever be considered to be fully inactive. For many allosteric proteins, including channels, there is a finite active probability even in the fully resting condition, with that probability influenced by the VSD equilibrium or ligand binding equilibrium.

This remark is well taken. Accordingly, we have addressed the exchange activity at rest. At a voltage of -30 mV and in the presence of 500 μ M amiloride, we exchanged the extracellular solution from symmetric to asymmetric, thus changing conditions from no net exchange to a net exchange, if the exchanger would be active. Under these conditions, we didn't observe an obvious change in pH_i . However, a subsequent voltage step to -100 mV elicited the respective change in pH_i . Obviously, the sensitivity of our recording system is limited and we cannot state that there is no basal activity at all. However, we estimate that it is below 3 % of its maximum value. We provide the additional data now in Fig S7A and B.

Specific Comments:

1. *Abstract: "cAMP directly modulates the voltage dependence of gating". Although the meaning of the authors is clear, more exactly it seems that cAMP is shifting the range of activation by voltage.*

We agree and have changed the wording accordingly.

2. *Abstract wording of last line. "endows the ... exchanger in sperm to produce a rapid" would be better as "endows the ... exchange with the ability to produce a rapid"*

We have changed the wording accordingly.

3. *p. 3 middle "mouse SLC9C1 is non-functional in heterologous systems" . Were attempts to measure gating currents done? I gather than the pH fluorimetry experiments should have shown something similar to what is reported in the present manuscript, if mouse SLC9C1 were functional.*

Wang and coworkers proposed in 2003 and 2007 that mouse SLC9C1 is non-functional in heterologous systems. However, the particular clone used in these studies lacked the first 55

amino acids, which incorporates the first transmembrane domain. This issue might have hampered the functional expression. Our own work using the full-length mouse SLC9C1 clone is only preliminary. However, the present status is that we don't measure any gating currents, indicating that either trafficking is impaired or that only few mouse SLC9C1 proteins make it into the plasma membrane.

4. p. 3 ... suggest “which serves as a switch to activate”

Wording has been corrected.

5. Last sentence of Intro suggestion: “our results now enable future study of the commonalities and”

We adopted the suggested wording.

6. p. 4. Title of first section. Comparison of putative protein topology isn't really structure, is it?

We changed the title accordingly.

7. Figure 2. The gating currents are much larger in the +cAMP example. Is that simply cell variability or a consistent finding?

This is an important question. The expression level of *Sp*SLC9C1 in cells varies considerably. Mean values of gating charges under different conditions are not different in the presence of cyclic nucleotides. We have included a summary of the gating charges recorded for the stable cell line and for transiently transfected cells (t) that we can include as supplementary information if requested (see below). The stable cell line had roughly 3 times more functional exchangers than the cells transiently expressing SLC9C1. We emphasize that none of our conclusions are dependent on the expression level of SLC9C1.

	“off” gating charges	Standard deviation
WT w/o	1.96196	1.07231
WT cAMP	1.74421	0.78852
WT cGMP	1.71895	0.99706
WT w/o (t)	0.61995	0.34251
R1053Q w/o (t)	0.65745	0.63229
R1053Q cAMP (t)	0.54641	0.1604
R803Q w/o (t)	0.46939	0.16592
R803Q cAMP (t)	0.65713	0.27972

8. *Figure 3. Although panels A and B show the Na concentrations in the experiment, I think it would also be useful to have the specific cytosolic and extracellular pH listed in the legends, as is done for Fig. 3C. This would also be helpful for Figure 4 and the supplementary figures, where appropriate.*

We now include the pH values in Figs. 3 and 4.

9. *Figure 3A vs. 3B. Why does the alkalization begin to fall after return to -40 mV in A, but the acidification persists in B? Is there some other pathway by which protons can enter the COS cells in A or might this be basal NHE activity? In both A and B, at -40 mV it is presumed there is no basal Na/H exchange at -40 mV, but that really isn't known, is it? In B, there is no extracellular Na, so NHE counted remove protons from the cytosol anyhow. Might the slow loss of alkalization in A reflect some very low rate of basal NHE activity even at -40 mV.*

We observe some cell-to-cell variability regarding the extent by which the pH value recovers after SLC9C1 activation. Using amiloride, we demonstrated that CHO cells harbor endogenous SLC9A1 exchangers. Their activity could underlie the recovery after SLC9C1 activation. However, the voltage dependence of pH changes was similar in the absence and presence of amiloride, demonstrating that the presence of endogenous exchangers does not compromise the results. Basal activity of SLC9C1 in part (A) would further alkalize the cell, given the ion gradients present in the cell.

10. *Is the much larger size of the acidification signal in B (relative to alkalization in A) significant? Is there some sidedness to the net flux rate?*

The size of the pH change that we record depends on the expression level of SLC9C1, cell size and geometry, and dye loading. We do not observe consistently higher pH changes in the reverse mode compared to the forward mode of the exchanger. However, we note that the ion gradients in the two recording configurations are not identical; in the forward mode, the Na⁺ gradient is about tenfold (140 vs. 14 mM), whereas in the reverse mode, the extracellular Na⁺ is completely removed (0 vs. 14 mM) creating a larger Na⁺ gradient.

11. *Figure 3C. I just want to suggest that this important panel does NOT show that exchange is not occurring, but that there is no NET proton flux.*

We agree and have rephrased the figure legend.

12. *Figure 3F-G-H, Figure 4A,C. The duration of each hyperpolarization leading to an acidification signal used to measure effect seems too short to reach a steady-state. Thus, the mean ΔR vs. voltage plots may be skewed by the fact that true steady-state is not measured. When comparing to cAMP this is not a problem, if the rates of NHE activation are similar, but that might not be the case. I don't think this is a serious problem, but it does seem curious that measurements were apparently made prior to steady-state.*

Indeed, the pH changes do not reach steady-state in our recordings, but even if they would, we couldn't state that we recorded the exchanger in steady-state. The intracellular pH is an indirect measure of SLC9C1 activity and we are confident that the slope of the pH changes is the appropriate measure to analyze exchanger activity. It is required to use the dye in its linear range; longer stimulations of SLC9C1 activity may saturate the response of the dye, and if pH values level off, the slopes no more provide information about exchanger activity.

13. *On this topic, Figure S3C indicates that once an Hv1 expressing COS cell is made alkaline, at least over the time course of these experiments, there is no mechanism to low pH in the cells. Some comment somewhere about the intrinsic voltage-dependence of Hv1 might be helpful to readers. My understanding is that Hv begins to activate around -50 mV. I think the authors want to use Fig. S3C to infer that changes in pH signals in cells coexpressing Hv1 and spSLC during steps between -23 and -113 mV won't be impacted by Hv1. But I'm not sure that is fully justified. At those potentials (-33 to -58 mV, Fig 3G) where no response is seen in the absence of cNMP, might the ongoing activation of Hv1 over this range mute any ability to see an NHE effect.*

The activation threshold of Hv1 depends strongly on the pH gradient. For the pH values used here ($pH_o = 7.4$, $pH_i = 7.2$), the threshold for voltage activation of Hv1 is ~ 0 mV. Only if the cell is strongly acidified by prolonged exchanger activation in the reverse mode, the voltage threshold of Hv1 would be shifted to more negative values. However, to study the voltage dependence, we always start the recording with activation of Hv1, resulting in the opposite effect, i.e. alkalization shifts the voltage dependence to more positive values. During SLC9C1 activation, except for three experiments, the intracellular pH was never more acidic than the starting pH value that we observed. The resulting mean values of SpSLC9C1 activation are not significantly

affected, irrespective of including or excluding these three experiments. Therefore, we are confident that the voltage dependence of Hv1 and SLC9C1 do not overlap.

14. *Is there a relatively specific inhibitor of Hv1 that could be used to test whether Hv1 activity might be compromising signals at more positive voltages? Overall, I don't see how any issues with such details impacts on the basic conclusions regarding effects of cAMP, but it just seems that some details of the methodologies could benefit from more description.*

There are Hv1 inhibitors that could be used for such an experiment, i.e. Zn²⁺ and 2GBI. However, our results are not compromised by Hv1 activity (see comments above), therefore, we think that the suggested experiment is not required.

15. *figure 4C. It might be useful to double to vertical scale on this panel, so the amplitude of the signals are more like in panel A.*

We agree and doubled the vertical scale.

16. *top of page 8. In saying that coexpression with Hv1 overcomes these limitation, it might be useful to readers to explain why that is.*

We explain now in more detail the rationale underlying the use of Hv1.

17. *Regarding the explanation of Figure S3C in the text, it is stated that stepping to negative values from +47 mV did not change pH. I think this may leave an incorrect impression. I presume that pH did not change, since there was mechanism available to reacidify the cells. And the absence of a pH change may be taken to indicate that Hv1 will not influence pH signals over those more negative voltages. However, Hv1 gating is altered in this range, so if there is a simultaneous process altering pH, then Hv1 gating in that range may impact on the pH signals, making measurements of spSLC9C1 activity not as quantitative as one might like.*

The experiment of Figure S3C allows two conclusions. First, Hv1 doesn't carry any inward H⁺ currents except tail currents during closure (see for example Sasaki et al., 2006, Science), that's why we chose that strategy. Second, within the time range of the pulse protocol, no re-acidification occurs by endogenous H⁺ transport systems. The activation threshold of Hv1 gating will be shifted for strong acidifications, however, going through all traces that we recorded, an acidification beyond the starting pH was only observed in three experiments. The resulting mean values are not significantly affected, irrespective of including or excluding these experiments. Therefore, our results are not compromised by Hv1 activity.

18. *bottom of p. 9. "CNBD modulates VSD movement" This may or may not be correct. I think what the data reveal are that the CNBD influences the VSD equilibrium. That probably has nothing to do with movement per se.*

We rephrased the sentence accordingly.

19. *Fig. S6C. It would be useful to indicate that the data with amiloride are without cAMP.*

This information is now provided in the Figure legend.

20. *2nd paragraph on p. 12. This touches on an issue already raised above. The question is, is activation of the VSD obligatory for any exchange to occur and the authors seem to assume the answer is affirmative. Physiological, that assumption may be largely correct, but mechanistically it will be an interesting question to be addressed in the future.*

We agree and include now a sentence that mentions the possibility that SLC9C1 – like ion channels – may exist in equilibrium between active and inactive states and that there might be a small activity at rest.

21. *p. 13. “During their lifetime sperm are exposed” particular sperm or all sperm? ‘unless triggered by a voltage pulse” Do sperm experience rapid voltage pulses?*

We clarified this issue and added a few sentences. The issue here is to distinguish between slower changes during spawning and rapid changes, while navigating on loop-like trajectories. This certainly applies to *Arbacia* sperm, but probably not to mammalian sperm navigating in the oviduct.

22. *near bottom. “This mechanism enables CatSper translating rapid...” change to “enables CatSper to translate rapid”*

Changed.

Response to referee #2:

1. *For Western blotting and immunofluorescence experiments (Fig. 6), the use of knock-out animal is the best control. This is almost impossible with sea urchin spermatozoa. At least, some control experiments to verify the antibody specificity should be performed with the primary antibody in the presence of excess amount of the antigen peptide. Signals from sperm head could be non-specific and control experiment with antigen peptide could resolve this issue.*

The unequivocal localization is an important issue. The reviewer correctly cautions that sperm heads are often non-specifically labelled by antibodies and, using excess antigen, specific staining could be prevented. However, there are two reasons, why we did not perform the requested experiment. First, antigen peptides may just as well remove unspecific labeling. Second, we also observe head and flagella localization by Western blotting of purified flagella and heads. Thus, whatever the outcome of the suggested experiment in the immunocytochemistry was, we could still not rule out the presence of the *SpSLC9C1* in sperm heads. The presence of SLC9C1 in *Strongylocentrotus purpuratus* sperm was demonstrated previously by mass spectrometry (Nomura & Vacquier, 2006, Cell Motility and the Cytoskeleton). We have now included a sentence in the result part referring to this work.

2. *The authors claimed that multiple band observed in Western blots may be due to posttranslational modifications. Is there previous evidence of such modification for other Na^+/H^+ exchangers? Do these modifications may alter their function?*

The SLC9A1 is O- and N-glycosylated, but its function does not depend on these modifications (Liu et al. BBA 2015, Fliegel et al. Biochemie 1994). Quite generally, most members of the SLC family are glycosylated (for a comprehensive review see Pedersen et al. Eur. J. Physiol. 2016). For some SLCs, glycosylation affects intracellular transport, protein stability and localization, but not function *per se* (“...mutation in N-glycosylation sites often fails to reveal consequences for transporter function”; Pedersen et al. 2016). We have not studied what kind of posttranslational modifications are occurring in *SpSLC9C1* and what impact on function they may have. Future studies are necessary to answer these questions.

3. *Endogenous Na^+/H^+ exchangers could be observed with this technique under different experimental and stimulating conditions?*

We're not sure whether we understand this question correctly. Endogenous exchangers were present in our cells as demonstrated by classical acid-load experiments (Figure S6). However, the voltage protocol used to characterize *SpSLC9C1* was also used in control cells, and endogenous exchangers are not activated by this voltage protocol. Also, we could demonstrate that the voltage dependence of *SpSLC9C1* activation is unchanged when endogenous exchanger are inhibited by amiloride. Therefore, we are confident that our results reflect *SpSLC9C1* properties only.

4. *Figures 3E and 3F show that the run down of *SpSLC9C1* depends on the test pulse (or ionic compositions of the experimental media). The authors should comment about an explanation for this phenomenon?*

“Run-down” is a common phenomenon often observed during prolonged recordings of ion channels. The reasons or mechanisms leading to “run-down” are in most cases not known. Phosphorylation, oxidation, the depletion of certain lipids, and the washout of important cofactors have been discussed as possible reasons for “run-down” of ion channels. We do not know the reason for run-down of *SpSLC9C1*. All experiments were completed before run-down became noticeable. To be utmost quantitative, we, therefore, used Hv1 co-expression to cut down on recording time.

5. *From the text it is not quite clear if SpSLC9C1 and hHv1 were co-expressed in all the experiments with CHO K1 cells expressing SpSLC9C1 in patch-clamp fluorometry. It is indicated in Methods that stable cell lines (CHO K1) were produced by electroporation of pc3 sNHE-HA or pc3 hHv1 (but not the combination). This should be clarified in Methods*

Co-expression of *SpSLC9C1* and Hv1 was only used to determine the voltage dependence of transport activity using BCECF. For all other experiments, cells only expressed *SpSLC9C1* alone. We clarified this now in the M & M section.

6. *In general, absolute R values of BCECF (F480/F440) are more informative to compare absolute pHi values (at least the difference between cells non-expressing and expressing SpSLC9C1). However, only delta R values instead of R values were used in all figures including Fig. S3. The reason to use delta R instead of R should be mentioned in the text.*

We emphasize that our pH recordings do not reflect absolute pH values. It is important to work in linear range of the dye, because then, we can normalize the ΔR values and determine mean values across experiments (see also the response to referee #1). Our estimate is that the dye is in its linear range at R values from 3.4 to 5.8. After dye loading and under reverse-mode conditions, cells typically displayed R values around 4.0 (see below). After Hv1 activation, R values increased to values between 4.5 and 5. From there, the voltage protocol to determine the voltage dependence was applied. We are prepared to include the plot of R values (see below) to the supplemental section, if requested.

7. *Authors utilized the null point method to calibrate pH. These experiments indicated that pHi value and potency of cytoplasmic pH buffer in the resting condition was not altered by the expression of SpSLC9C1. This is a quite important conclusion. However, the experimental procedure is not described in detail and the cited references are not sufficient to understand the procedure. Detailed experimental procedure should be described in the text.*

We thank the reviewer for pointing this out. We added explanatory sentences to the M & M section and to the legend of Fig. S3. Other than that, the null-point method is described in detail in M & M and the result section of ref. 9, but also in the original papers.

8. *Immunofluorescence signal of SpSLC9C1 in HEK K1 cells shows non-homogenous distribution (forming some patches) in the plasma membrane (Fig S2). The gating current indicates that SpSLC9C1 was really expressed in the plasma membrane without question, but some comments would be required for this unusual expression pattern.*

Such a staining pattern is not unusual. Staining patterns often depend on various small technical details. It doesn't necessarily reflect the real distribution of the antigen. We attribute no further importance to this staining, and no conclusion rests on this staining pattern other than the protein is made in the cell.

9. *SpSLC9C1 has an additional TM in the N-terminus compared to mammalian SLC9C1 (Fig. S1). If the authors have some insights about this difference related to the successful expression, it should be mentioned in the text.*

We have no further insight whether TM1 facilitated expression compared to mammalian SLC9C1. Of note, the sequence published by Wang et al. 2003 is missing the first 55 aa that encompass TM1. Thus, this clone was incomplete. We don't know whether this missing part was responsible for the failure to become functionally expressed (see also the response to referee #1).

10. *To establish the specificity of the CNBD the authors should do a concentration study and not use a practically saturating concentration (1 mM) of the compounds. The endogenous cAMP is significantly lower under physiological conditions so the authors should discuss if the exchanger is regulated at lower concentrations.*

For several reasons a quantitative dose-response relation is difficult if not impossible to establish. First, it is impossible to establish in the whole-cell mode a voltage dependence \pm cAMP. Second, at lower concentrations, the true cAMP concentration is not known, because of PDE activity. That's why we used 1 mM cAMP, a presumably saturating concentration not compromised by PDE activity. However, the experiment using caged cAMP indicates that $V_{1/2}$ becomes shifted at cAMP concentrations much lower than 1 mM. In addition, for sperm recordings, we used a concentration of 30 μ M DEACM-cAMP, a compound that does not accumulate in cells. We obtained robust responses uncaging cAMP this way. Therefore, the affinity of the exchanger for cAMP is likely to be in the micromolar range.

11. Not being an expert in NHE, I consulted some of the given references. I noticed that references 14 and 18 (from the authors), are cited to highlight the importance of pH in chemoattraction, but actually these papers were reported to minimize the role of pH_i changes in sea urchin sperm physiology. For example, the abstract of reference 18 states “We conclude that an elevation of pH_i is required neither to open Ca²⁺-permeable channels nor to control the chemotactic behavior”. Another example is in the discussion section, with the sentence: “The alkalization primes CatSper channels to open and promotes Ca²⁺ entry”, the work cited by the authors (9) demonstrates the presence of CatSper in sea urchin sperm but it does not demonstrate its fundamental participation in chemotaxis as huge concentrations of nonspecific blockers were used to inhibit chemotaxis. The authors should at least cite other evidence supporting the key role of CatSper in chemotaxis. This imprecise use of references is misleading and contributes to a distortion in the way readers perceive the development of the field and an unfair attitude in Biological Sciences, it should be corrected.

We had no intention to present a biased reference list. Science has progressed over the years and our work on CatSper channels (ref. 9) shows that pH_i changes are more important than previously thought. The exquisite pH sensitivity of CatSper (ref. 9) provides a mechanistic rationale why pH changes are important. For work on pH_i in sea urchin sperm, we now include a paper and a review from a different group to the reference list (González-Cota et al., 2015, *FEBS Lett.*; Nishigaki et al., 2014, *Biochem Biophys Res Commun.*).

We take issue with the reviewer’s statement that, in a previous publication, we used “huge concentrations of non-specific blockers”. Because CatSper channels have only been identified recently, its pharmacology is still developing. In ref. 9, we have used two compounds that feature completely different chemical structures (Mibefradil, MDL12330A). Both compounds blocked calcium signals in *Arbacia punctulata* sperm. We used MDL12330 (10 μM) for our assay of chemotaxis. The concentration we used was chosen according to the K_i of the drug, which is, depending on the procedure to activate CatSper, between 6 and 16 μM. Of note, 10 μM of the drug completely abolished chemotaxis. Thus, the concentration is moderate to low with respect to K_i. In addition, we scrutinized the literature for other evidence that supports the involvement of CatSper in chemotaxis. There are two other reports (Espinal-Enriquez et al. 2017 *Sci.Rep.* and Rennhack et al., 2018, *Brit. J. Pharmacology*, *in press*). Espinal-Enriquez and co-workers demonstrate that CatSper channels are present in *S. purpuratus* sperm. Although, no experiments regarding chemotaxis were performed, we now cite Espinal-Enriquez et al. 2017. Rennhack et al. characterize a more specific CatSper blocker and demonstrate that it inhibits chemotaxis-induced Ca²⁺ signals and chemotaxis of *A. punctulata* sperm. The paper will be online within the next few weeks, we will be happy to include Rennhack et al. 2018 in our reference list.

12. Minor point:

In page 11 line 4, “resact-evoked” should be “speract-evoked”

Corrected.

Response to referee #3

1. The major claim of the paper that SLC9C1, Na⁺/H⁺ exchanger is also operated by voltage is somehow predictable. For someone very familiar with ion channels can predict a large aspect of the study. The cyclic nucleotide binding domain (CNBD) a structure that binds cAMP and cGMP is known for years as an important regulator for several ion channels. The structure and the operation mode of the voltage sensor domain (VSD) in ion channels are much known. The authors used mutations that are known to either affect CNBD of the VSD domain. Although the data are of interest to others in the community, the finding is not novel per se. I should give credit though to the authors for this interesting and nice characterisation that was somehow expected to come out. I have few suggestions for a full characterization.

We take issue with the notion that our results were predictable and not novel *per se*: there are several ion channels that carry a VSD, but are not voltage-gated, and there are several ion channels and PKAs that carry a CNBD, but are not activated by cyclic nucleotides. We state this *expressis verbis* on page 9 of the manuscript. To emphasize this issue even further, we include a similar sentence regarding the presence of a VSD in voltage-independent channels.

2. Although the data are of interest to others in the community, the finding is not novel per se. Wang, D., et al., King, S.M., Quill, T.A., Doolittle, L.K. & Garbers, D.L. A new spermspecific Na⁺/H⁺ exchanger required for sperm motility and fertility. Nat. Cell Biol. 5, 1117-1122 (2003). Wang, D. et al. A sperm-specific Na⁺/H⁺ exchanger (sNHE) is critical for expression and in vivo bicarbonate regulation of the soluble adenylyl cyclase (sAC). Proc. Natl.Acad. Sci. USA 104, 9325-9330 (2007).

We disagree. These studies had a completely different focus and lack a functional characterization of mouse SLC9C1. In fact, the authors worked with a partial clone lacking the first 55 amino acids. The only functional data that was presented was that of a chimera, where the reportedly first transmembrane domain (in fact, it was the second one) was replaced by the first three transmembrane domains of an SLC9A exchanger. The logic of this experiment is unclear; moreover the functionality of that construct is minor if at all. In conclusion, all of our functional characterization is *novel per se*.

3. The V_{1/2} value of the off-gating current is very shifted to negative voltages! This means that at a resting membrane potential of -50mV of a sperm the voltage sensor is permanently in ON position. Question, what is the resting membrane potential of an Urchin Sperm? This is important, to have an idea on the state of the VSD and therefore the Na⁺/H⁺ activity at rest.

Voltage dependent movement of gating charges and transport activity of SpSLC9C1 of SpSLC9C1 depend on the concentrations of cAMP. In the absence of cAMP, at V_{rest} of -50 mV, the exchanger is mostly inactive. However, when cAMP is elevated during chemotactic signaling, the voltage dependence of charge movement and that of Na⁺/H⁺ exchange are both shifted to more positive voltages. We changed the wording to avoid any misunderstanding. The resting voltage V_{rest} is about -50 mV, as the reviewer states correctly.

4. Another point what ion channel is responsible of changing voltage in sperm? Can this or these channels be inhibited? This important to show the physiological role on the SLC9A.

The sperm cell is hyperpolarized by opening of a K⁺-selective CNG channel (Strünker et al. *Nat. Cell Biol.* 2006; Galindo et al., 2007 *Biochem. Biophys. Res. Commun.*; Bönigk et al., 2009, *Sci. Signal.*). When the opening of this CNGK channel is prevented, the changes in pH_i are also prevented (see e.g. Seifert et al. 2015, *EMBO J.*). The physiological role of SLC9C1 is shown and described throughout Fig. 5.

5. Delete the VSD and CNBD on SLC9C1 and study the Na⁺/H⁺ properties and vice versa attach a VSD or/and CNBD to a mammalian Na⁺/H⁺ and study if this regulated by these added structures.

We trust that the mutations that we studied revealed the regulation of *Sp*SLC9C1 by the respective domains. For a further characterization, we prefer to obtain a 3D structure of *Sp*SLC9C1 by high-resolution EM.

REVIEWERS' COMMENTS:

Reviewer #1 (Remarks to the Author):

I am comfortable with the changes made by the authors and feel this is an important contribution.

Reviewer #2 (Remarks to the Author):

I have reviewed the rebuttal letter by F. Windler and colleagues regarding their paper: "The solute carrier SLC9C1 is a Na⁺/H⁺ exchanger gated by an S4-type voltage sensor and cyclic nucleotide binding". I consider that the authors addressed all the observations/suggestions raised by all reviewers. In few instances they did not, but they provided a well justified reason/argument to defend their original idea/conclusion, etc. Therefore, I recommend publication of the revised manuscript that has improved considerably with the modifications.

Reviewer #3 (Remarks to the Author):

As I stated in my first review this is a very nice characterization of SLC9C1 Na⁺/H⁺ exchanger . The first report of the finding of this gene was published and studied several years ago (Wang, D., et al., King, S.M., Quill, T.A., Doolittle, L.K. & Garbers, D.L. A new spermspecific Na⁺/H⁺ exchanger required for sperm motility and fertility. Nat. Cell Biol. 5, 1117-1122 (2003). Wang, D. et al. A sperm-specific Na⁺/H⁺ exchanger (sNHE) is critical for expression and in vivo bicarbonate regulation of the soluble adenylyl cyclase (sAC). Proc. Natl.Acad. Sci. USA 104, 9325-9330 (2007)). The authors claimed that here are several ion channels that carry a VSD, but are not voltage-gated. However, the authors are not stating that 95% if not more of the proteins that exhibit a voltage-sensor domain are regulated by voltage.